# Learning to Explore: Scaling Agentic Reasoning via Exploration-Aware Policy Optimization

Xingyuan Hua [1]  Sheng Yue [2]  Ju Ren [1 3]

## Abstract

Recent advancements in agentic test-time scaling allow models to gather environmental feedback before committing to final actions. A key limitation of existing methods is that they typically employ undifferentiated exploration strategies, lacking the ability to adaptively distinguish when exploration is truly required. In this paper, we propose an exploration-aware reinforcement learning framework that enables LLM agents to adaptively explore only when uncertainty is high. Our method introduces a fine-grained reward function via variational inference that explicitly evaluates exploratory actions by estimating their potential to improve future decision-making, together with an exploration-aware grouping mechanism that separates exploratory actions from task-completion actions during optimization. By targeting informational gaps, this design allows agents to explore selectively and transition to execution as soon as the task context is clear. Empirically, we demonstrate that our approach achieves consistent improvements across a range of challenging text-based and GUI-based agent benchmarks. Code is available at `https://github.com/HansenHua/EAPO-ICML26` and models are available at `https://huggingface.co/hansenhua/EAPO-ICML26`.

## 1. Introduction

Recent advances in agentic models have demonstrated transformative impact across a large number of real-world domains (Wang et al., 2023c; Zheng et al., 2023; Yue et al., 2024a; Jimenez et al., 2024; Zhong et al., 2024; Chen et al., 2025a), where models can make decisions based on current states and interact with the environment. Yet, current agentic models often struggle in complex, long-horizon settings, such as web navigation (Yao et al., 2022a; Kong et al., 2025), scientific research (Yang et al., 2023; Rein et al., 2024), and embodied agentic tasks (Wang et al., 2023a; Song et al., 2023; Yue et al., 2024e), because their goal-oriented training objective easily limits the ability to generalize in unfamiliar scenarios and obtain environmental information for deeper reasoning (Krishnamurthy et al., 2024).

Very recently, research has shifted towards agent test-time scaling (Yao et al., 2023; Snell et al., 2024; Tajwar et al., 2024; Setlur et al., 2025). In this context, an agent can commit multiple candidate actions, receive the resulting feedback or environmental changes, and update its internal reasoning or plan accordingly (Yang et al., 2025c; Jiang et al., 2025). This process allows the agent to gather additional information about the environment or task dynamics before committing to a final action, effectively enabling adaptive, multi-step reasoning during deployment (Pathak et al., 2017; Yao et al., 2022b; Lee et al., 2025). Such paradigms are expected to improve reasoning and decision-making accuracy by enhancing the agent's understanding of the environment through additional contextual information, and have shown great potential across various complex agentic tasks, including mobile agent navigation (Rawles et al., 2025; Kong et al., 2025) and interactive web tasks (Yao et al., 2022a; Xie et al., 2024).

Albeit achieving improved performance, we find that current test-time scaling methods entangle exploration and action selection within a single policy, preventing agents from identifying where exploration is truly necessary and often resulting in indiscriminate exploration even in well-understood states. This conservative exploration strategy leads agents to accumulate low-value information and obscure the most critical signals. In contrast, humans naturally separate information-seeking exploration from final decision making by assessing which parts of the environment are uncertain and selectively performing exploration to resolve these uncertainties (Wilson et al., 2014). This separation becomes particularly advantageous when agents encounter

---

[1]Department of Computer Science and Technology, Tsinghua University, Beijing, China [2]School of Cyber Science and Technology, Sun Yat-sen University Shenzhen Campus, Shenzhen, China [3]State Key Laboratory of Internet Architecture, Tsinghua University, Beijing, China. Correspondence to: Sheng Yue <yuesh5@mail.sysu.edu.cn>.

*Proceedings of the 43rd International Conference on Machine Learning*, Seoul, South Korea. PMLR 306, 2026. Copyright 2026 by the author(s).

| Method | Qwen-VL-2B | Qwen-VL-4B | Qwen-VL-8B |
|--------|-----------|-----------|-----------|
| GRPO | $52.3 \rightarrow 55.1$ | $56.1 \rightarrow 58.4$ | $\mathbf{55.4 \rightarrow 52.9}$ |
| DAPO | $\mathbf{56.8 \rightarrow 54.6}$ | $60.6 \rightarrow 58.5$ | $61.7 \rightarrow 63.1$ |
| GiGPO | $\mathbf{56.0 \rightarrow 53.1}$ | $59.7 \rightarrow 61.8$ | $58.8 \rightarrow 56.7$ |
| LAMER | $\mathbf{61.2 \rightarrow 56.3}$ | $\mathbf{61.5 \rightarrow 58.2}$ | $\mathbf{61.0 \rightarrow 57.3}$ |

*Table 1.* Performance of naive exploration on AndroidWorld. We demonstrate the performance shifts and highlight the cases with degradation.

unfamiliar states that deviate from the training distribution. By making exploration an explicit process, agents can leverage distributional mismatch as a signal to guide information acquisition at test time.

Drawing inspiration from humans' adaptive exploration paradigm, we seek to answer: *"How can agentic models explore at the appropriate state to obtain adequate information for decision making?"* A straightforward solution is to instruct the agent to try alternative actions when facing an unfamiliar state until sufficient information is gathered. However, as illustrated in Table 1, current methods fail to fully benefit from exploration as they lack the ability to pursue valuable actions and incorporate explored information (Krishnamurthy et al., 2024). Instead, a more reasonable approach is to teach agents to distinguish when exploration is informative and when direct goal-pursuit is sufficient, enabling them to adaptively allocate interaction steps based on uncertainty rather than relying on ad-hoc prompting. Although promising, it is highly challenging to evaluate the utility of exploratory actions and balance exploration with exploitation.

To tackle these challenges, we propose an *exploration-aware policy optimization* (EAPO) method for efficient agent learning, which teaches agents to explore at proper states, capable of allowing agents to make attempts and obtain dynamic information at test-time. First, we introduce an exploration-and-memory reasoning mode that allows the agent to explicitly generate exploration guidance and summarize newly observed states, thereby making exploratory behavior an integral part of the reasoning process. To accurately characterize the utility of actions, we further train a reward function that enables the agent to distinguish when exploration is necessary and how exploratory actions can benefit subsequent decision-making, effectively mitigating overly conservative behaviors. Furthermore, we develop an exploration-aware two-stage training strategy, including SFT rollback and exploration-aware GRPO, leading to more stable and effective optimization.

We systematically evaluate the proposed method across 4 challenging environments, including embodied agentic tasks, online shopping tasks, and web/mobile GUI control. The results demonstrate that EAPO significantly enhances decision-making capability across all environments, consis-

tently outperforming existing methods by 20%–60%, particularly in complex long-horizon GUI control tasks. Further, EAPO incurs only about 30% additional training overhead while enabling a 2B-scale model to outperform most substantially larger general and agentic models. In addition, we observe that agents exhibit adaptive exploration behavior at test time and can generalize directly to unseen scenarios without requiring additional fine-tuning.

**Conflict of Interest Disclosure.** We declare that there are no financial or other substantive conflicts of interest related to this work.

## 2. Related Work

**LLM Test-Time Scaling.** Recent advances in large language models (LLMs) have stimulated growing interest in test-time scaling (Snell et al., 2024) for reasoning and decision-making beyond single-step generation (Wei et al., 2022; Madaan et al., 2023; Wang et al., 2025b). Several works utilize prompting strategies to branch multiple reasoning trajectories and select or aggregate final answers, thereby diversifying intermediate reasoning paths during inference (Yao et al., 2022b; Du et al., 2023; Yao et al., 2023; Wang et al., 2023b; Shinn et al., 2023; Besta et al., 2024; Liao et al., 2025; Hua et al., 2026). (Yao et al., 2023) introduce an inference framework to improve long-horizon thinking capability by considering and self-evaluating multiple different reasoning paths for the final decision. (Tian et al., 2024) integrates Monte Carlo Tree Search (MCTS) with LLMs to establish a self-improving loop, thereby enhancing the capabilities of LLMs without additional annotations. However, these approaches largely rely on static heuristics or predefined branching budgets and lack principled criteria to adaptively control when and how exploration should be conducted during multi-step reasoning. More recently, researches have attempted to overcome this limitation by utilizing entropy as an extra signal to balance exploration and exploitation during multi-step reasoning (Zhang et al., 2024; 2025a; Vanlioglu, 2025; Xu et al., 2025). (Zhang et al., 2025a) utilize entropy to dynamically adjust the exploration depth during multi-step reasoning. (Vanlioglu, 2025) introduce entropy into advantage estimation to enable efficient exploration while maintaining training stability. However, entropy does not faithfully reflect information gain, as actions with high entropy may simply indicate model uncertainty while inducing uninformative or redundant transitions, thus failing to produce meaningful exploration.

**RL for LLM Agents.** Reinforcement learning (RL) has recently attracted significant attention (Yue et al., 2024d;c), as it encourages the exploration of diverse reasoning chains under the guidance of verifiable rewards (Yue et al., 2024b; Ahmadian et al., 2024; Yu et al., 2025; Zheng et al., 2025;

Lu et al., 2025; Feng et al., 2025). One line of work focuses on balancing exploration and exploitation during policy optimization, encouraging diverse action selection and preventing premature convergence (Zhang et al., 2024; 2025a; Vanlioglu, 2025; Xu et al., 2025). However, such solutions primarily enhance exploration during the training phase, rather than enabling agents to perform explicit and adaptive exploration at test time when interacting with unfamiliar states. Beyond exploration during training, some recent methods propose to enhance agent robustness through explicit exploration or refinement mechanisms at test-time (Tajwar et al., 2024; Gandhi et al., 2024; Setlur et al., 2025; Zhang et al., 2025c;b). (Jiang et al., 2025) propose a general meta-RL framework that enables LLM agents to first explore several trajectories and learn from the environment feedback at test time. (Yang et al., 2025c) select the best action proposal from multiple candidates to expand searching space and improve planning robustness. Albeit with promising results, these methods tend to induce overly conservative behaviors by applying exploration or refinement uniformly across all situations, rather than enabling agents to reason about when exploration is necessary, thereby limiting their effectiveness in adaptive decision-making.

## 3. Preliminaries

**Agentic Tasks.** We frame the agentic tasks as an MDP, $\langle \mathcal{S}, \mathcal{A}, P, T, R, \mu, \gamma \rangle$, with $\mathcal{S}$ the state space, $\mathcal{A}$ the action space, $P$ the environment dynamics, $T$ the episodic horizon, $R$ the reward function, $\mu$ the initial state distribution, and $\gamma$ the discount factor. $P(s'|s, a)$ represents the probability of transitioning to state $s'$ after taking action $a$ in state $s$. An agentic model $\pi_\theta(a|s)$, parameterized by $\theta \in \mathbb{R}^d$, defines a distribution over actions conditioned on the current state. Rolling out $\pi_\theta$ with the environment induces a trajectory, $\tau = \{s_1, a_1, s_2, a_2, \ldots, s_T\}$, whose likelihood is given by $p(\tau|\theta) \doteq \mu(s_1) \prod_{t=1}^{T-1} P(s_{t+1}|s_t, a_t)\pi_\theta(a_t|s_t)$. The learning objective is to maximize the expected discounted cumulative reward:

$$\max_\theta J(\theta) \doteq \mathbb{E}_{\tau \sim p(\cdot|\theta)} \left[ \sum_{t=1}^{T-1} \gamma^t R(s_t, a_t, s_{t+1}) \right]. \quad (1)$$

Consider GUI-based agentic tasks (Hong et al., 2024; Li et al., 2025). Here, $\mathcal{S}$ corresponds to all possible visual UI contexts paired with task descriptions, $\mathcal{A}$ includes executable actions such as tapping or swiping at specific screen coordinates or entering text, and $P$ captures the underlying navigation logic of the application. The horizon $T$ specifies the maximum number of interaction steps. The agent's behavior is typically governed by an LLM policy $\pi_\theta$. At each step $t$, the agent observes the current UI states and generates a textual action $a_t = (w_1, w_2, \ldots, w_n) \in \mathcal{V}^n$, where each $w_i$ is a token from the vocabulary $\mathcal{V}$. The action is then

parsed into an executable command. The reward function $R$ is often a sparse, binary success signal indicating if the agent completes the task.

**Group Relative Policy Optimization (GRPO).** GRPO is a practical method widely used for training agentic models (Shao et al., 2024). This method generates a group of trajectories with the same task description $g$ and concatenates all the generated tokens of the $i$-th trajectory into a complete action, $a_i = \{a_{i,1}, a_{i,2}, \ldots, a_{i,|a_i|}\}$. Then, the training objective is defined as:

$$J(\theta) = \mathbb{E}_{g \sim \mu, a_{i=1:G} \sim \pi_{\text{old}}(\cdot|g)} \left[ \frac{1}{G} \sum_{i=1}^{G} \frac{1}{|a_i|} \sum_{t=1}^{|a_i|} \min \left\{ w_{i,t}\tilde{A}_{i,t} \right. \right.$$
$$\left. \left. , \text{clip}(w_{i,t}, 1-\epsilon, 1+\epsilon)\tilde{A}_{i,t} \right\} - \lambda \text{KL}(\pi_\theta \| \pi_{\text{ref}}) \right] \quad (2)$$

where $G$ is the number of generated trajectories in each group, and $\lambda$ is a hyperparameter. The importance weight $w_{i,t}$ and advantage $\tilde{A}_{i,t}$ of token $a_{i,t}$ are defined as:

$$w_{i,t} = \frac{\pi_\theta(a_{i,t}|g, a_{i,<t})}{\pi_{\text{old}}(a_{i,t}|g, a_{i,<t})} \quad (3)$$

$$\tilde{A}_{i,t} = \frac{R(g, a_{i,t}) - \text{mean}(\{R(g, a_{i,t})\}_{i=1}^{G})}{\text{std}(\{R(g, a_{i,t})\}_{i=1}^{G})} \quad (4)$$

where $R(g, a_{i,t})$ represents the reward-to-go of $a_{i,t}$.

## 4. Exploration and Memory Mode

### 4.1. Motivation

Owing to the multi-turn, interactive nature of agentic tasks and potentially out-of-distribution environments (such as updated UI layouts in web navigation or unmapped topologies in robotic pathfinding) during execution, it is of great importance to endow the agentic model with the ability to proactively explore the environment and memorize historical viewed states during execution (Jiang et al., 2025). Drawing inspiration from the success of OpenAI-o1 (Jaech et al., 2024) and DeepSeek-R1 (Guo et al., 2025b) in test-time compute, we next extend the test-time scaling beyond pure logical reasoning to active exploration and memorization: equipping the agentic model with structured exploration guidance and an explicit memory that summarizes previously visited states into a persistent log.

Let $e_t$ denote a structured exploration strategy that specifies what information the agent needs to acquire next and what candidate actions it needs to take to achieve this goal. Let $m_t$ represent an accumulative summary of task-relevant information extracted from past interactions. Set the initial exploration cue $e_0$ and memory $m_0$ as empty strings. Then,

at each execution step $t$, besides the task description $g$ and current state $s_t$, we introduce the preceding exploration strategy $e_{t-1}$ and memory $m_{t-1}$ into the input of the agentic model. Accordingly, the output consists of not only the executable action $a_t$ but also the current exploration $e_t$ and accumulative memory $m_t$. Formally, we have:

$$\tilde{a}_t = \pi_\theta(\cdot|\tilde{s}_t), \qquad (5)$$

where $\tilde{s}_t \doteq [g; s_t; e_{t-1}; m_{t-1}]$ and $\tilde{a}_t \doteq [e_t; m_t; a_t]$.

By incorporating exploration and accumulative memory into dedicated fields, the agent has to reason about the requisite exploration and the synthesis of acquired environmental information. It is expected to enhance the agentic model's "atomic capability" in retrieving informative past interactions and actively understanding environments.

*Remark* 4.1. Of note, this structured format does not constrain the semantic content of exploration or memory, but only organizes them into dedicated fields. This separation helps the agent explicitly distinguish between decision-making, information gathering, and state summarization, which facilitates more effective reuse of acquired information and reduces ambiguity during optimization. In particular, the format enables reliable credit assignment for exploration-related rewards, which would otherwise be difficult to achieve with unstructured outputs.

### 4.2. Instruction Template

To operationalize the explicit modeling of exploration and memory, it is necessary to generate outputs that strictly follow a predefined format. Inspired by Chen et al. (2025b), we introduce the `<explore>` and `<memory>` tags as additional components in the agent's output (see detailed instructions in Fig. 5). The `<explore>` tag is used to capture candidate actions and intermediate environmental probes, allowing the agent to deliberate on (potentially unfamiliar) environmental dynamics before committing to an execution. The `<memory>` tag distills previously visited states and acquired information into a structured summary, serving as an externalized working memory that can be referenced across multiple decision steps (detailed in Appendix C.5).

### 4.3. Reward Modeling

Directly applying the above instruction during training is insufficient to incentivize the agent to explore uncertainty and organize memories, as the standard learning objectives relying on success/failure signals essentially encourage the agent to learn the reactive mapping between states and optimal actions of the training tasks. To tackle this issue, we next design a fine-grained reward model to explicitly credit the valuable exploratory and mnemonic behaviors.

The central challenge is how to accurately quantify the util-

ity of the exploratory actions. A straightforward solution would be to estimate the exploratory (trial-and-error) actions via online rollouts to obtain the empirical returns. Yet, it faces a fundamental dilemma: a small sample size can induce high variance due to policy stochasticity, while scaling the number of rollouts incurs substantial computational and interaction costs (detailed in Appendix B).

Our key insight is that learning to explore is fundamentally the process of *training the agentic model to correctly enrich its memory* by proactively acquiring useful task-relevant information. We formalize this from a Bayesian perspective. Denote $p(e_{t-1}, m_{t-1}|s_t, \text{success})$ as the posterior exploration-memory distribution, conditioned on task success, which can characterizes the utility of specific exploration strategies and memory states in facilitating successful trajectories from state $s_t$. A higher probability indicates that the exploration-memory can provide requisite informational gain to resolve the environmental uncertainty for task completion. Leveraging this, for any transition sample $(\tilde{s}_t, \tilde{a}_t, \tilde{s}_{t+1})$, we define the Bayesian exploratory reward as:

$$R_{\text{explore}}(\tilde{s}_t, \tilde{a}_t, \tilde{s}_{t+1}) \doteq \max \Big\{ p(e_{t-1}, m_{t-1}|s_t, \text{success})$$
$$, \gamma^2 p(e_{t-1}, [m_{t-1}, s_{t+1}]|s_t, \text{success}) \Big\} \qquad (6)$$

where $\tilde{s}_t = [g; s_t; e_{t-1}; m_{t-1}]$ and $\tilde{a}_t = [e_t; m_t; a_t]$. Here, success indicates that the corresponding trajectory initiated from state $s_t$ completes the task.

*Remark* 4.2. In Eq. (6), the first term quantifies the utility of *immediate exploitation*, that is, using existing memory to solve the task. The second term characterizes the *proactive exploration*, where newly acquired information (the explored state $s_{t+1}$) is concatenated with the existing memory. The reward ensures that the actions leading to the states with valuable information for future decisions are properly credited, incentivizing the agent to "look ahead" and understand the environment whenever the current state $s_t$ is uncertain or the current memory $m_{t-1}$ is insufficient for task completion.

*Remark* 4.3. The discount factor $\gamma^2$ in Eq. (6) is essential for preventing *overconservatism*. Once the agent has already acquired sufficient information to make a correct decision at the current state, continued exploration solely for obtaining more comprehensive memories is redundant and can be detrimental to efficiency (Shinn et al., 2023; Chen et al., 2024). Since the benefit of exploration is not immediate – requiring at least one step to observe a new state and a subsequent step to synthesize the information – we apply a $\gamma^2$ penalty to the exploratory gain. It guides the agent to carry out exploration only when the anticipated utility 'outweighs' the latency cost.

Since the true posterior $p(\cdot|\cdot, \text{success})$ is intractable, we ap-

proximate it using a learnable variational proxy $q_\phi(e, m|s)$, parameterized by $\phi$. We treat $q_\phi$ as a 'policy' that selects optimal exploration-memory configurations and train it to minimize the KL divergence with the true posterior:

$$\min_{q_\phi} \mathrm{KL}(q(e, m|s)\|p(e, m|s, \mathrm{success})). \qquad (7)$$

To optimize Eq. (7), we utilize variational inference to derive a surrogate objective (detailed in Appendix A):

$$\max_{q_\phi} \beta \mathbb{E}_{e, m \sim q_\phi(\cdot|s)}[Q(s, e, m)] - \mathrm{KL}(q(e, m|s)\|p(e, m|s))$$
$$(8)$$

where $\beta$ is a hyperparameter. From Eq. (7), we can optimize the variational distribution $q_\phi(e, m|s)$ using REINFORCE (Williams, 1992). More specifically, consider $q_\phi$ as a policy that selects $(e, m)$ conditioned on $s$. Then, the objective can be viewed as a KL-regularized policy optimization, where $Q(s, e, m)$ serves as the the cummulative reward of the trajectory starting from $s$ with $e, m$ generated by $q(\cdot|s)$ and action generated by policy $\pi(\cdot|s, e, m)$, and the KL term acts as a functional constrain preventing the learned proxy from deviating the prior and collapsing.

Therefore, the exploratory reward can be computed as:

$$R_{\mathrm{explore}}(\tilde{s}_t, \tilde{a}_t, \tilde{s}_{t+1}) = \max \Big\{ q_\phi(e_{t-1}, m_{t-1}|s_t)$$
$$, \gamma^2 q_\phi(e_{t-1}, [m_{t-1}, s_{t+1}]|s_t) \Big\}. \qquad (9)$$

The density-based reward in Eq. (9) provides a stable and efficient way to evaluate the utility of exploratory actions. By modeling a distribution over memories conditioned on the current state, the estimation of action utility is robust to the policy stochasticity. In addition, it decouples reward estimation from active environment interaction. It eliminates expensive online rollouts and remains scalable to large-scale training and complex environments.

*Remark* 4.4. Our exploratory reward $R_{\mathrm{explore}}$ is principled for maximizing task success rate. By treating memory and exploration as *latent variables* within the reasoning process, we show that Eq. (8) corresponds to a lower-bound log-likelihood objective for the success rate estimation (see Eq. (18) for details). Consequently, maximizing Eq. (8) directly promotes maximization of the task success rate.

Finally, the total reward for a transition $(\tilde{s}_t, \tilde{a}_t, \tilde{s}_{t+1})$ is a weighted combination of three modules: the exploratory reward $R_{\mathrm{exploration}}$, the format reward $R_{\mathrm{format}}$, and the success signal $R_{\mathrm{task}}$, i.e.,

$$R(\tilde{s}_t, \tilde{a}_t, \tilde{s}_{t+1}) \doteq R_{\mathrm{task}}(\tilde{s}_t, \tilde{a}_t, \tilde{s}_{t+1})$$
$$+ \alpha_1 R_{\mathrm{format}}(\tilde{s}_t, \tilde{a}_t, \tilde{s}_{t+1}) + \alpha_2 R_{\mathrm{explore}}(\tilde{s}_t, \tilde{a}_t, \tilde{s}_{t+1})$$
$$(10)$$

where $\alpha_1$ and $\alpha_2$ are hyperparameters. The format reward $R_{\mathrm{format}}$ is binary and determined by whether the output correctly follows the predefined structured templates (e.g., correct tags and \boxed{} actions, encouraging the model to give structured, parsable outputs. As in Section 3, $R_{\mathrm{task}}$ serves as a episodic binary reward, indicating whether the corresponding trajectory successfully reaches the task goal.

## 5. Exploration-Aware Training

While the proposed reward model provides an accurate characterization of action utility, its direct implementation for training agentic policies remains non-trivial. This is primarily because the estimated rewards cannot be reliably attributed to the correct decisions under standard training pipelines, leading to biased optimization signals. On the one hand, the current agentic model fundamentally lacks the "rollback" capability, that is, it lacks the functional awareness to autonomously return to a previous state (such as clicking a "back" button) once an action leads to an undesirable state. Without the rollback capability, the reward assigned to an exploratory action is entangled with its irreversible downstream consequences, preventing the agent from correctly attributing future success to the information gained through exploration. As a result, the utility of exploratory actions is underestimated.

On the other hand, when applying the policy optimization method like GRPO, exploratory actions and task-execution actions may be mixed within the same optimization groups. This grouping blurs the distinction between information-gathering and goal-executing behaviors, misleading relative advantage estimation, even when exploratory actions are essential for long-term task success. To tackle these challenges, we introduce EAPO, a practical two-stage training algorithm for exploration-aware agent training.

### 5.1. Learning to Rollback

We leverage Supervised Fine-Tuning (SFT) to train the agent to acquire the rollback capabilities. Specifically, we collect an expert rollback transition dataset $\mathcal{D}$ by prompting a teacher LLM with a state $s'$ and its previous state $s$ to generate the rollback action $a$. The transition $(s', s, a)$ will be admitted into $\mathcal{D}$ if the action can successfully recover the state. The agent is prompted with a rollback instruction $x$ then trained to minimize the loss:

$$\mathcal{L}_{\mathrm{SFT}}(\theta) = -\frac{1}{|\mathcal{D}|} \sum_{(s', s, a) \in \mathcal{D}} \log p_\theta(a|s', s, x). \qquad (11)$$

By SFT on these expert rollback transitions, the agent learns to reliably recover the previous state, allowing it to treat exploration not as a terminal risk, but as a reversible behavior.

## 5.2. Exploration-Aware Policy Optimization

After obtaining the ability to rollback, we proceed to optimize the agentic policy under the proposed reward. In principle, the complete state includes not only the environment state but also the agent's exploration information and memory, as these jointly determine which action is appropriate in Eq. (5). In practice, however, grouping transitions by this complete state is infeasible, since exploration histories and memories can differ substantially even when the environment state is identical, leading to an excessive number of distinct states and impractical advantage estimation.

Therefore, we introduce the *visitation depth*, denoted by $\kappa(s_t^i)$, as the number of times the agent revisits the same state during exploration. Formally, we have:

$$\kappa(s_t^i) \doteq \sum_{k<t} \mathbb{I}[s_k^i = s_t^i]. \tag{12}$$

We cluster the transitions based on their joint the environment states and the visitation depths. To be specific, we first generate a group of trajectories with the same initial state and task goal using the old policy $\pi_{\text{old}}$, denoted as $\tau_1, \ldots \tau_G$. Then, these transitions are clustered into localized transition groups $\mathcal{G}(s, \nu)$:

$$\mathcal{G}(s, \nu) \doteq \left\{ (\tilde{s}_t^i, \tilde{a}_t^i, \tilde{s}_{t+1}^i) \big| s_t^i = s, \kappa(s_t^i) = \nu, \right. \\ \left. 1 \leq i \leq G, 1 \leq t \leq T \right\}. \tag{13}$$

*Remark* 5.1. Clustering transitions with the visitation depth is crucial for accurate advantage estimation in exploration-aware agentic training. The return of exploratory actions is inherently lower than that of the final action executed after exploration due to discounting. Without distinguishing their exploration stages, directly comparing these actions within the same group underestimates the value of exploration, leading to a significant decrease in exploration degree (see details in Fig. 19). By incorporating visitation depth into the clustering criterion, actions are compared only with others at similar informational stages, which enables more faithful evaluation of their contributions and stabilizes the learning of exploratory behaviors.

Finally, we compute the advantage of the $j$-th token in action $\tilde{a}_t^i$ in group $\mathcal{G}(s_t^i, \kappa(s_t^i))$ by:

$$\tilde{A}_{i,j} = \frac{R(\tilde{s}_t^i, \tilde{a}_t^i, \tilde{s}_{t+1}^i) - \text{mean}(\{R(\tilde{s}_t^i, \tilde{a}_t^i, \tilde{s}_{t+1}^i)\}_{i \in \mathcal{G}})}{\text{std}(\{R(\tilde{s}_t^i, \tilde{a}_t^i, \tilde{s}_{t+1}^i)\}_{i \in \mathcal{G}})}. \tag{14}$$

Overall, we name our proposed algorithm EAPO, with its pseudocode detailed in Algorithm 1.

## 6. Experiment

In this section, we conduct experiments to evaluate the performance of EAPO by answering the following research questions:

- How does EAPO perform compared to existing methods and models across various benchmarks, especially in complex GUI-based agentic tasks?
- What are the effects of key parameters and components?
- How do agents learn exploration strategies and execute exploration at test-time?
- How does the learned model generalize to unseen environments?

### 6.1. Experimental Setup

**Environments.** We run experiments with 2 domains including 4 environments: **1) Text-based**, including ALFWorld (Shridhar et al., 2021) and WebShop (Yao et al., 2022a). **2) GUI-based**, including AndroidWorld (Rawles et al., 2025) and OSWorld (Xie et al., 2024). Detailed Description on environments can be found in Section C.1.

**Baselines.** We evaluate our method against six strong baseline methods: **1)** Min-p (Minh et al., 2025) adjusts the sampling threshold based on model confidence using the top token's probability as a scaling factor.. **2)** OverRIDE (Shi & Pan, 2026) dynamically adjusts the probability distribution of the next word to explore new and low-probability generation paths. **3)** GRPO (Shao et al., 2024) optimizes LLM policies by computing relative advantages within trajectory groups without training a critic. **4)** DAPO (Yu et al., 2025) improves long-chain-of-thought RL by dynamically sampling trajectories and using relaxed clipping to stabilize group-based optimization. **5)** GiGPO (Feng et al., 2025) extends group-based policy optimization with two-dimensional credit assignment across steps and trajectories for multi-turn learning. **6)** LAMER (Jiang et al., 2025) trains LLM agents to adapt online by sampling multiple trajectories and reasoning over them through in-context.

**Reproducibility.** All details of our experiments are provided in Section C.3 in terms of the tasks, network architectures, hyperparameters, etc. We conduct experiments on different model size suitable for real-world deployment, which are Qwen3 (Yang et al., 2025a) with 3 model sizes (1.7B, 4B, and 8B) for text-based environments and Qwen3-VL (Bai et al., 2025) with 3 model sizes (2B, 4B, and 8B) for GUI-based environments. Model deployment employs vLLM as the rollout service and the sampling temperature is set to 1, with a maximum generation length of 4096 tokens. All the experiments are run on Ubuntu 22.04.4 LTS with 8 NVIDIA H800 GPUs.

### 6.2. Experimental Results

**Comparative Results.** To answer the first question, we evaluate EAPO's performance across all datasets, with vary-

| Model | ALFworld | WebShop | AndroidWorld | OSWorld |
|---|---|---|---|---|
| *Closed-source Models* | | | | |
| OpenAI CUA o3 (OpenAI, 2025) | 42.31 | 38.73 | 55.43 | 23.00 |
| TianXi-Action-7B (Tian et al., 2024) | 36.57 | 32.46 | 48.93 | 29.81 |
| DeepMiner-Mano-72B (Fu et al., 2025) | 54.12 | 49.71 | 68.48 | 53.88 |
| Seed1.5-VL-250717 (Guo et al., 2025a) | 47.53 | 43.09 | 61.75 | 40.18 |
| UI-TARS-2-2509 (Wang et al., 2025a) | 51.21 | 46.85 | 73.38 | 53.11 |
| Claude-4-Sonnet-0929 (Anthropic, 2025b) | 56.09 | 50.97 | 71.60 | 62.88 |
| *Open-source Models* | | | | |
| Qwen3-VL-235B-A22B (Bai et al., 2025) | 50.81 | 45.05 | 62.00 | 38.10 |
| ZeroGUI (Yang et al., 2025b) | 35.76 | 31.29 | 47.52 | 20.20 |
| UI-TARS-7B (Qin et al., 2025) | 31.82 | 28.55 | 33.04 | 27.52 |
| OpenCUA-32B (Wang et al., 2025c) | 39.99 | 35.48 | 51.66 | 34.79 |
| ARPO (Lu et al., 2025) | 37.20 | 33.17 | 49.31 | 29.90 |
| GUI-Owl-7B (Ye et al., 2025) | 38.47 | 34.06 | 52.04 | 34.79 |
| DART-GUI-7B (Li et al., 2025) | 45.78 | 40.83 | 57.99 | 42.13 |
| *Training Methods* | | | | |
| GRPO (Shao et al., 2024) | 46.13 | 38.57 | 55.48 | 40.36 |
| DAPO (Yu et al., 2025) | 51.86 | 40.90 | 61.76 | 47.90 |
| GiGPO (Feng et al., 2025) | 56.76 | 42.24 | 58.87 | 50.88 |
| LAMER (Jiang et al., 2025) | 61.26 | 47.74 | 60.98 | 55.60 |
| *Ours* | | | | |
| EAPO-1.7B/2B | 58.50 ↓2.76 | 53.28 ↑2.31 | 76.36 ↑1.98 | 50.34 ↓12.54 |
| EAPO-4B | 69.00 ↑7.74 | 60.84 ↑9.87 | 79.59 ↑6.21 | 57.89 ↑4.99 |
| EAPO-8B | 76.02 ↑14.76 | 65.58 ↑14.61 | 82.05 ↑8.67 | 64.29 ↑1.41 |

*Table 2.* Success rate using different agentic models on text-based environments. We exhibit the performance advantage with the best baseline and highlight the  best  result.

ing base models. We present selected results in Table 2 and Fig. 1. Full comparisons with current strong general and agentic models are reported in Tables 4 and 5, while detailed comparisons with baseline methods are provided in Figs. 2 and 9. We find EAPO consistently outperforms baselines in all 4 environments, often by a significant margin in terms of performance and convergence. Notably, a 2B-scale model trained with EAPO achieves higher performance than substantially larger general and agentic models, demonstrating that EAPO effectively enables agents to learn when and how to explore, thereby substantially improving environment understanding and decision quality during execution. ToT (Yao et al., 2023) performs explicit test-time exploration by branching multiple reasoning paths and selecting actions via self-evaluation, but its exploration is purely inference-driven and lacks a learning signal, resulting in uniform and often inefficient exploration across states. In contrast, LAMER (Jiang et al., 2025) enables agents to explore by sampling multiple trajectories and adapting behavior through in-context meta-learning; however, its exploration is applied uniformly and tends to be conservative, as it does not explicitly learn when exploration is necessary. Compared to both, our method explicitly learns exploration-aware policies through reward modeling and grouping mechanisms, allowing agents to reason about

when and how to explore, thereby achieving more efficient and adaptive exploration during execution.

**Key Parameters.** To answer the second question, we conduct experiments with varying the discount factor (ranging from 0.5 to 1.0), sampling group size (ranging from 4 to 32), and KL coefficient (ranging from 0.005 to 1.0). The data and parameter setup adhere to that of Table 3. We present full results in Figs. 10 to 12 of Sections D.4 to D.6. We observe that increasing $\gamma$ generally promotes exploration by preserving rewards obtained through information-gathering actions, but overly large values may lead to excessive exploration and introduce irrelevant information that degrades final decision quality. A larger group size $G$ improves performance up to a point by providing more reliable relative advantage estimates, while excessively large groups incur diminishing returns due to reduced update frequency and higher variance across trajectories. Similarly, the KL coefficient $\lambda$ exhibits a unimodal effect: moderate values stabilize training and improve performance by preventing overly aggressive updates, whereas overly small or large $\lambda$ either lead to unstable optimization or overly constrained policies that hinder effective exploration. Therefore, it is crucial to appropriately adjust these hyperparameters to balance exploration and exploitation, thereby achieving stable optimization and

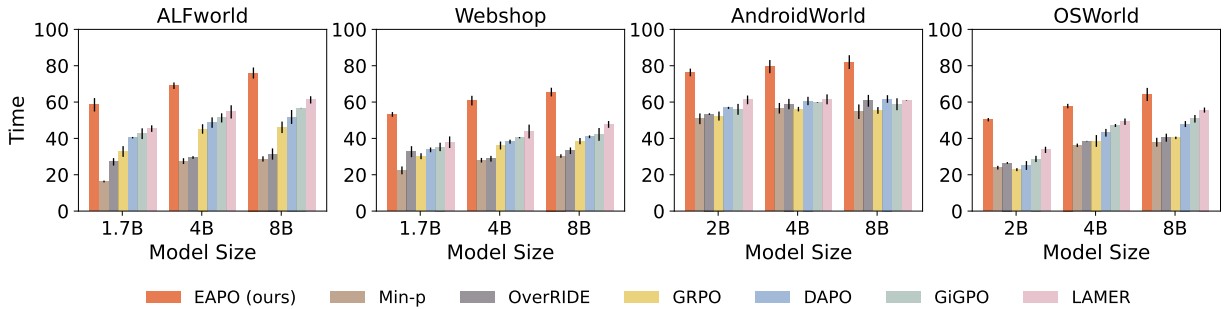

*Figure 1.* Performance with varying model size. Uncertainty intervals depict standard deviation over three seeds. EAPO exhibits higher performance, demonstrating the effectiveness of exploration during test time and its great efficiency in encouraging agents to explore compared to existing methods.

strong task performance across diverse environments.

**Online Reward.** To validate the efficiency of the proposed reward model, we conduct experiments using an alternative online exploratory reward as a comparison, which samples trajectories to estimate the utility of memory online. We present the full results in Figs. 3 and 4 of Section B. We gradually increase the number of online samples used for reward estimation (ranging from 1 to 10) and observe that performance consistently improves with more samples. Notably, our method achieves performance comparable to the online reward with a large number of samples, demonstrating that the learned reward model can accurately evaluate the value of actions and memory while avoiding the costly overhead of extensive online sampling.

**Group Size.** To answer the third question, we verify the group size distribution during updates. Full results are shown in Fig. 16 of Section D.7.4. The results clearly show that, in the early stages of training, the frequency of larger step-level groups increases rapidly, reflecting that the agent increasingly revisits states and actively explores to acquire additional information. As training progresses, the distribution gradually stabilizes, suggesting that the agent learns to distinguish when exploration is necessary and avoids indiscriminate exploration, thereby achieving a more balanced and effective exploration–exploitation behavior.

**Exploration Degree.** To further answer the third question, we demonstrate the exploration degree during training. We present full results in Fig. 17 of Section D.7.5. As shown in the results, the exploration degree initially increases as the agent learns to actively visit informative states and acquire additional environmental information, indicating the emergence of effective exploratory behavior. As training proceeds, the exploration degree gradually converges, suggesting that the agent learns to selectively explore only when necessary rather than engaging in indiscriminate exploration. This adaptive exploration strategy efficiently overcomes the tendency of being overly conservative, leading to better generalization capability in unseen scenarios.

**Case Studies.** We present a case in OSWorld to demonstrate how agent adaptively explore at test time. Full results are demonstrated in Section E. During inference, our model adaptively explores uncertain environments by outputting candidate actions in the <explore>mode. It then executes these actions, observes the resulting states, summarizes them, and outputs the memories into <memory>mode. We further observe that the agent may perform multi-step exploration along a certain direction. Through this process, the agent accumulates knowledge about state transitions, enabling it to make more informed and effective decisions in subsequent steps. We present the full trajectories in OSWorld in https://github.com/HansenHua/EAPO-ICML26.

**Ablation Studies.** We assess the effect of key components by ablating them on all datasets under the same setting. *1) Importance of exploratory reward. 2) Importance of exploration-aware grouping. 3) Importance of format reward.* As illustrated in Table 8 and Figs. 18 and 19, removing *exploratory reward* causes the agent to fail to evaluate the usefulness of each exploratory action, causing ineffective exploration and premature convergence to suboptimal behaviors, which leads to significantly degraded performance across all environments. Without *exploration-aware grouping*, exploratory and task-execution actions are mixed within the same optimization groups, resulting in exploratory actions being increasingly underestimated as training progresses; this induces a rise-then-collapse pattern in both exploration degree and task performance, indicating unstable exploration and impaired long-term optimization. Without the *format reward*, the agent fails to consistently follow the predefined output structure for exploration signals and memory updates, preventing effective organization, storage, and reuse of information obtained through exploration, and thereby limiting the agent's ability to leverage exploratory behaviors to improve decision-making and task completion.

**Generalization.** To answer the fourth question, we conduct experiments on applying models trained on Android-

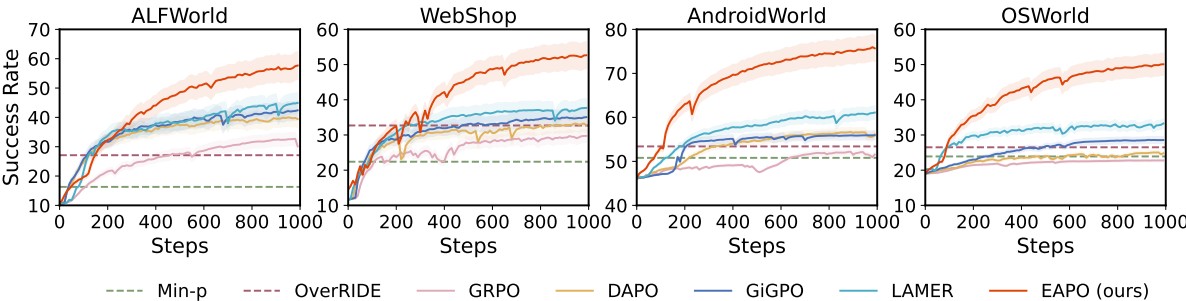

*Figure 2.* Training convergence with varying model size. Uncertainty intervals depict standard deviation over three seeds. EAPO consistently and significantly surpasses existing methods in terms of convergence speed and stability.

World on unseen environments (OSWorld) and present the full results in Section D.3. We attribute this generalization primarily to the explicit modeling of exploration and memory at test time. By disentangling exploratory reasoning from action execution and maintaining a structured memory of previously visited states, the agent is able to adapt its interaction strategy to new domains without requiring environment-specific retraining. As a result, the learned exploration policy exhibits domain-invariant characteristics, enabling effective generalization to previously unseen tasks and applications.

**Run Time.** To demonstrate the training efficiency of our method, we verify its runtime across all the environments and present full results in Figs. 20 and 21. EAPO incurs less than a $15\%$ increase in training time compared to existing methods. This overhead mainly stems from training the variational distribution. Notably, the additional cost is negligible when compared to the expense of online trajectory sampling, which is commonly required by alternative approaches. Given the substantial improvements in performance and convergence speed, this modest runtime overhead is a reasonable and acceptable trade-off.

As for inference time, we present comparison of the average step in Section D.7.7 and observe that our method indeed has additional runtime due to exploration. To avoid meanless exploration, we introduce introduce a discount factor $\gamma$ so that our method maintains only a modest increase (at most 10) in average step. To be specific, we apply a discount to the exploratory gain since the benefit of exploration is not immediate – requiring at least one step to observe a new state and a subsequent step to synthesize the information It guides the agent to carry out exploration only when the anticipated utility 'outweighs' the latency cost, which will avoid meanless exploration even when agent has obtained enough information.

## 7. Limitation and Discussion

In this paper, we propose a novel exploration-aware policy optimization method that teaches agents to explore at

appropriate states and effectively leverage the acquired information for decision-making. By explicitly modeling exploration utility and incorporating exploration-aware optimization, EAPO enables agents to distinguish when exploration is beneficial and when it should be avoided. Extensive experiments on agentic tasks corroborate the effectiveness of exploration at test-time and superiority of EAPO. A limitation of EAPO lies in its reliance on structured exploration and memory representations, which are manually specified throughout training. This may restrict the expressiveness of exploration strategies and limit adaptability to tasks that require more flexible forms of information acquisition.

## Acknowledgement

This research was supported in part by the National Natural Science Foundation of China under Grants 62572496 and 62432004, the Fundamental and Interdisciplinary Disciplines Breakthrough Plan of the Ministry of Education of China under Grant No. JYB2025XDXM122, the Guangdong Natural Science Foundation under Grant 2026A1515011265, the Shenzhen Science and Technology Program under Grant JCYJ20250604175500001, the Young Elite Scientist Sponsorship Program by CAST under Contract ZB2025-218, a grant from the Shenzhen Science and Technology Program under Grant No. SYSRD20250529113401002, a grant from the Guoqiang Institute, Tsinghua University, and the Smartchip Research Project of the Chinese Institute of Electronics under Grant No. 2024-02.

## Impact Statement

This work advances the understanding and design of exploration mechanisms for agentic large language models by introducing a principled framework that enables agents to learn when and how to explore. However, the broader implications of deploying such models warrant careful consideration. More capable exploration may increase agent autonomy and effectiveness in complex environments, which, if misused or insufficiently constrained, could lead to unintended behaviors or amplified risks in real-world systems.

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

## A. Derivation

We aim to find a memory distribution $q(e, m|s)$, which is closest to the original distribution $p(e, m|s, a)$. Formally, the objective is defined as:

$$\min_q \mathrm{KL}(q(e, m|s) \| p(e, m|s, \text{success})). \tag{15}$$

Based on the definition of KL divergence, we can derive:

$$
\begin{aligned}
\mathrm{KL}(q\|p) &= \mathbb{E}_q[\log q(e, m|s) - \log p(e, m|s, \text{success})] \\
&= \mathbb{E}_q[\log q(e, m|s) - \log p(\text{success}|s, e, m) - \log p(e, m|s) + \log p(R|s)] \\
&= -\mathbb{E}_q[\log p(\text{success}|s, e, m)] + \mathrm{KL}(q(e, m|s)\|p(e, m|s)) + \log p(\text{success}|s). 
\end{aligned} \tag{16}
$$

Since $\log p(R|s)$ is irrelevant to memory, we can derive an alternative objective for minimizing Eq. (15):

$$\max_q \mathbb{E}_q[\log p(\text{success}|s, e, m)] - \mathrm{KL}(q(e, m|s)\|p(e, m|s)). \tag{17}$$

We can also prove that this objective serves as the lower bound to maximizing the probability that the agent $\pi_\theta$ generates a success trajectory starting from state $s$ as $\pi_\theta(\text{success}|s)$. Based on ELBO (Kingma & Welling, 2013), we treat memory and exploration as latent variables and derive a variational lower bound:

$$
\begin{aligned}
\log \pi_\theta(\text{success}|s) &= \log \sum_{m,e} \pi_\theta(\text{success}|s, m, e)\pi_\theta(m, e|s) \\
&= \log \sum_{m,e} \pi_\theta(\text{success}|s, m, e)\frac{\pi_\theta(m, e|s)}{q_\phi(m, e|s)}q_\phi(m, e|s) \\
&= \log \mathbb{E}_q[\pi_\theta(\text{success}|s, m, e)\frac{\pi_\theta(m, e|s)}{q_\phi(m, e|s)}] \\
&\geq \mathbb{E}_q[\log \pi_\theta(\text{success}|s, m, e)\frac{\pi_\theta(m, e|s)}{q_\phi(m, e|s)}] \\
&= \mathbb{E}_q[\log \pi_\theta(\text{success}|s, m, e)] - \mathrm{KL}(q(m, e|s)\|\pi_\theta(m, e|s)). 
\end{aligned} \tag{18}
$$

Therefore, we can derive an alternative objective:

$$\max_q \mathbb{E}_q[\log \pi_\theta(\text{success}|s, m, e)] - \mathrm{KL}(q(m, e|s)\|\pi_\theta(m, e|s)). \tag{19}$$

For the first term $p(\text{success}|s, e, m)$, we apply the law of total probability over all possible trajectories:

$$p(\text{success}|s, e, m) = \int_\tau p(\text{success}|\tau)p(\tau|s, e, m)d\tau. \tag{20}$$

According to the regularized soft policy, the optimal exploration and memory policy can be expressed as $q(e, m|s) \propto \pi_{\text{ref}}(e, m|s)\exp(Q(s, e, m))$, where $Q(s, e, m) = \sum_t r_t$ denotes the cummulative reward of the trajectory starting from $s$ with $e, m$ generated by $q(\cdot|s)$ and action generated by policy $\pi_\theta(\cdot|s, e, m)$. Then, we can derive;

$$p(\tau) \propto p(s)\prod_t p(s_{t+1}|s_t, e_t, m_t)\pi_{\text{ref}}(e_t, m_t|s_t)\exp(Q(s_t, e_t, m_t)) \tag{21}$$

Based on (Levine, 2018), we can derive:

$$p(\text{success}, \tau) \propto p(s)\prod_t p(s_{t+1}|s_t, e_t, m_t)\exp(r(s_t, e_t, m_t)) \tag{22}$$

Based on the definition of Bayesian conditional probability, we can derive:

$$p(\text{success}|\tau) = p(\text{success}, \tau)/p(\tau)$$

$$\propto \frac{p(s) \prod_t p(s_{t+1}|s_t, e_t, m_t) \exp(r(s_t, e_t, m_t))}{p(s) \prod_t p(s_{t+1}|s_t, e_t, m_t)\pi_{\text{ref}}(e, m|s) \exp(Q(s_t, e_t, m_t))}$$

$$= \frac{p(s) \exp(Q(s, e, m)) \prod_t p(s_{t+1}|s_t, e_t, m_t)}{p(s) \prod_t p(s_{t+1}|s_t, e_t, m_t)\pi_{\text{ref}}(e, m|s) \exp(Q(s_t, e_t, m_t))}$$

$$\propto \exp(Q(s, e, m)). \tag{23}$$

Here, the derivation of the last line is due to the sparse binary reward in agentic tasks, which indicates that the Q-function equals the reward at the last round $Q(s_t, e_t, m_t) = r_T$.

Substituting in Eq. (20), we can derive:

$$p(\text{success}|s, e, m) \propto \int_\tau \exp\left(\sum_t r_t\right) p(\tau|s, e, m) d\tau$$

$$= \mathbb{E}_{\tau \sim \pi(\cdot|s,e,m)}\left[\exp\left(\sum_t r_t\right)\right], \tag{24}$$

where $\pi(\cdot|s, e, m)$ denotes the policy for generating action.

Using the Jensen inequality, we can derive:

$$\log p(\text{success}|s, e, m) = \beta \log \mathbb{E}_{\tau \sim \pi(\cdot|s,e,m)}\left[\exp\left(\sum_t r_t\right)\right]$$

$$\geq \beta \mathbb{E}_{\tau \sim \pi(\cdot|s,e,m)}\left[\sum_t r_t\right], \tag{25}$$

where $\beta$ is a hyperparameter.

Substituting in Eq. (17), we derive the final objective:

$$\max_q \beta \mathbb{E}_{e,m \sim q(\cdot|s)}[Q(s, e, m)] - \text{KL}(q(e, m|s)\|p(e, m|s)). \tag{26}$$

## B. Alternative Reward Function

In this section, we provide an alternative online exploratory reward. This exploratory reward characterizes the utility of an action from two perspectives. The first part $R_1$ is determined by the direct rollout obtained after committing to the action $a_t$. Formally, we define $R_1$ as:

$$R_1(\tilde{s}_t, \tilde{a}_t) \doteq \sum_{i=t}^T \gamma^{i-t} r(s_i, a_i). \tag{27}$$

This assigns a high reward to correct actions that move the agent closer to the target state, which encourages goal-directed behavior and efficient task completion when sufficient information is already available.

Unlike existing approaches which underevaluate the actions that are not immediately correct but informative for decision-making. The second part $R_2$ evaluates the refined rollout obtained when the agent is allowed to explore future states. To be specific, the agent transits to $s_{t+1}$ and rolls back to $s_t$ with action $a_r$. Then, the agent generates a refined action $a'_t$, exploration guidance $e'_t$, and memory $m'_t$, denoted as:

$$a'_t, e'_t, m'_t = \pi_\theta(\cdot|g, s_t, e_{t-1}, [m_{t-1}, s_{t+1}]) \tag{28}$$

This refined action influences the subsequent decision process, yielding another trajectory starting from $s_t$, denoted as $\tau' = \{s_t, a_t, s_{t+1}, a_r, s_t, a'_t, \ldots, s'_H\}$. Formally, we define $R_2$ as:

$$R_2(\tilde{s}_t, \tilde{a}_t) \doteq r(s_t, a_t) + \gamma r(s_{t+1}, a_r) + \sum_{i=t}^T \gamma^{i-t+2} r(s'_i, a'_i). \tag{29}$$

Then, the exploratory reward function is defined as:

$$R_{\text{explore}} \doteq \max\{R_1, R_2\}. \tag{30}$$

We further verify the exploration degree of the online reward. As illustrated in Fig. 3, the exploration degree is slightly lower than the trained reward. The underlying reason is that this online reward may underestimate the value of exploratory actions, as current policy may struggle to accurately capture the long-term information gain of exploratory actions.

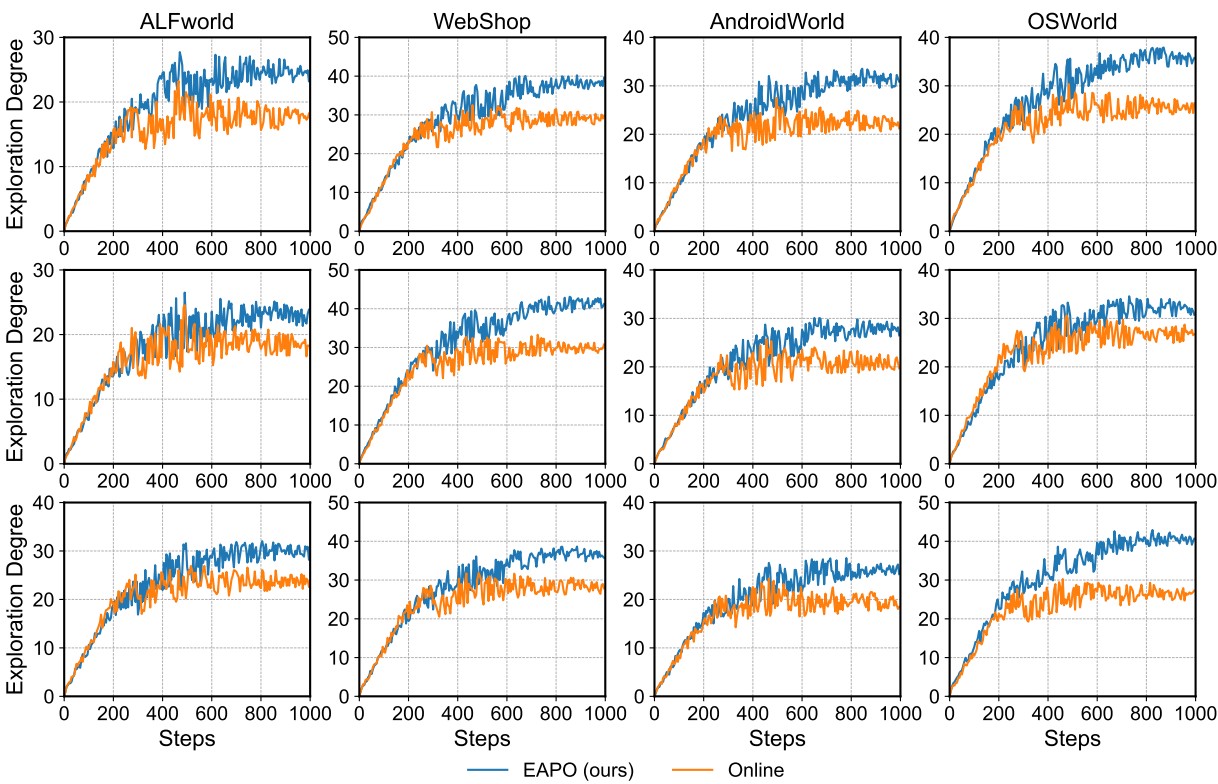

*Figure 3.* Exploration degree comparison between EAPO and the alternative online reward.

To validate that sampling more trajectories can reduce this underestimation, we conduct experiment by varying the number of trajectories, ranging from 1 to 10. As illustrated in Fig. 4, performance consistently improves as the number of trajectories increases, indicating that more accurate estimation of action utility provides stronger supervision for policy optimization. Notably, our method achieves performance comparable to that of multi-trajectory sampling, demonstrating that the proposed reward model can effectively mitigate estimation inaccuracies without requiring expensive online sampling.

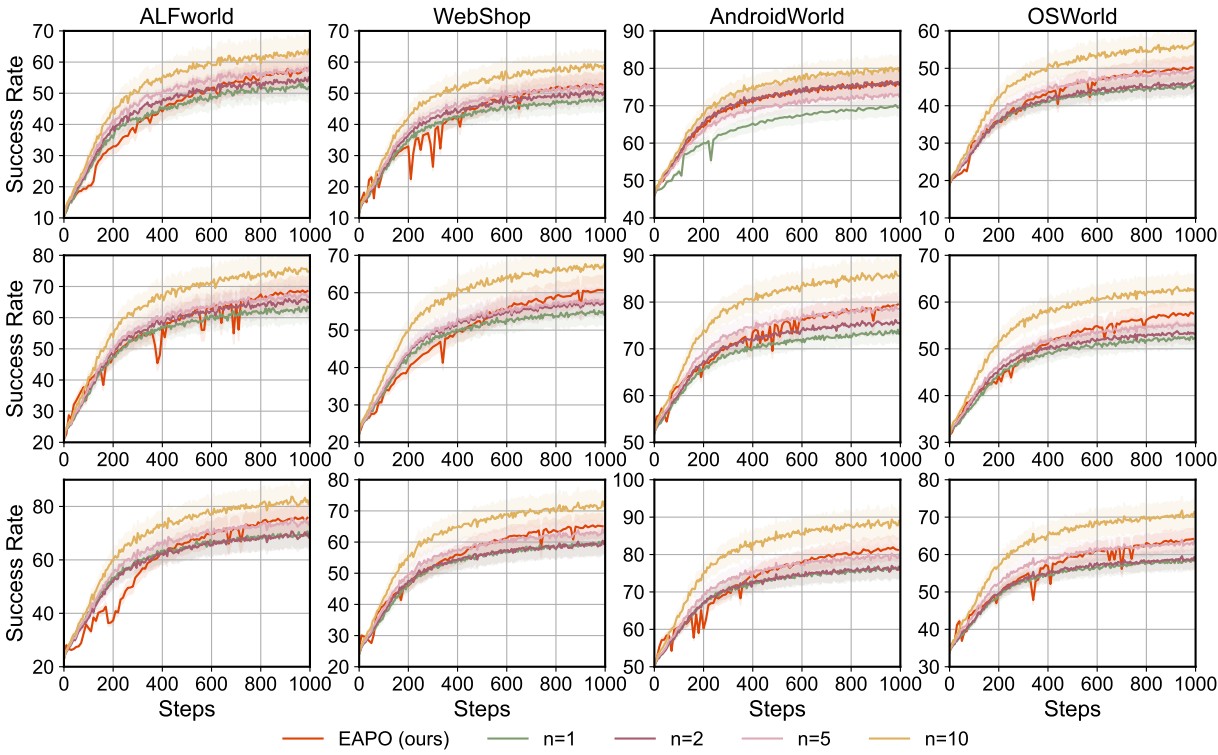

*Figure 4.* Exploration degree comparison between EAPO and the alternative online reward.

# C. Experiment Setup

## C.1. Environments

We evaluate our method on two areas with 4 environments, which are widely used in prior studies. We elaborate on what follows.

- **ALFworld** (Shridhar et al., 2021), a text-based embodied environment featuring household tasks, where agents navigate and interact with objects via natural language commands.
- **WebShop** (Yao et al., 2022a), a complex, web-based interactive environment designed to test the LLM agents in realistic online shopping scenarios, requiring the agent to explore and plan under uncertainty to finish the task.
- **AndroidWorld** (Rawles et al., 2025), an environment with 116 dynamic tasks across 20 real-world Android apps, designed to evaluate mobile agents' capabilities in app navigation and system-level control.
- **OSWorld** (Xie et al., 2024), a large-scale web benchmark with 369 tasks that span real-world web and desktop applications, requiring long-horizon planning and multi-window tool coordination.

## C.2. Baselines

We test our method against six baselines. We implement them based on their publicly available implementations.

- Min-p (Minh et al., 2025), a dynamic truncation method that adjusts the sampling threshold based on model confidence using the top token's probability as a scaling factor.
- OverRIDE (Shi & Pan, 2026), a decoding method that dynamically adjusts the probability distribution of the next word in the inference stage, encouraging the model to explore new and low-probability generation paths.
- Group Relative Policy Optimization (GRPO) (Shao et al., 2024), a group-based, critic-free reinforcement learning method that estimates advantages over trajectory groups, providing stable optimization for reasoning-oriented LLM training.
- Dynamic Sampling Policy Optimization (DAPO) (Yu et al., 2025), a group-based, critic-free RL approach that improves training efficiency and stability in long chain-of-thought settings through higher clipping thresholds and dynamic sampling.
- Group in Group Policy Optimization (GiGPO) (Feng et al., 2025), a group-based RL algorithm that introduces two-dimensional credit assignment across steps and trajectories, making it suitable for multi-turn optimization of LLM agents.
- LLM Agent with Meta-RL (LAMER) (Jiang et al., 2025), a meta-reinforcement learning framework that allows LLM agents to sample multiple trajectories and adapt their behavior through in-context interaction with these sampled experiences.

## C.3. Implementation Details

The reward function is implemented by the same model as the policy model for all the environments. We provide a detailed hyperparameter setting in Table 3.

*Table 3.* Hyperparameters (identical across datasets).

| Hyperparameter | Value |
|---|---|
| Number of RL epochs | 1000 |
| Sampling group size | 16 |
| Weight of format reward $\alpha_1$ | 0.5 |
| Weight of exploratory reward $\alpha_2$ | 1 |
| Weight of Discount factor $\gamma$ | 0.9 |
| Learning rate of reward model | 1e-4 |
| Learning rate of policy model | 1e-4 |
| KL loss coefficency $\lambda$ | 0.01 |

We implement our code using Pytorch 2.8.0, built upon the open-source framework of verl (Sheng et al., 2024), provided at `https://github.com/volcengine/verl`. All the experiments are run on Ubuntu 22.04.4 LTS with 8 NVIDIA H800 GPUs.

## C.4. Pseudocode

We present the Pseudocode of EAPO in Algorithm 1.

---

**Algorithm 1** Pseudocode of EAPO

---

1: Prepare the rollback dataset $\mathcal{D}$ and initial reward model $q_\phi(m|s)$, policy network $\pi_\theta(a|s, m)$.
2: **for** each SFT step **do**
3:     Sample transition data $(s, a) \sim \mathcal{D}$ and update the policy network via the loss function Eq. (11).
4: **end for**
5: **for** each RL step **do**
6:     **for** $i = 1$ to $G$ **do**
7:         Sample trajectory $\tau_i$ based on Eq. (5).
8:     **end for**
9:     Group trajectories into state-action transition via Eq. (13).
10:     Optimize the reward model via the objective Eq. (8)
11:     Obtain the reward by Eq. (9) and Eq. (10).
12:     Estimate the advantage of each action in the transition groups via Eq. (14) and update the policy network.
13: **end for**

---

## C.5. Instruction Template

The instruction templates used in our agentic system consist of three components: *system prompt* in Fig. 5, *basic prompt* in Fig. 6, and *action guidance* in Fig. 7. The system prompt specifies the high-level cognitive process that the agent should follow, including how it reasons about the environment, conducts exploration, and maintains intermediate memory. This prompt will be input in system prompt of instruction-tuned models and remains fixed across different tasks. The basic prompt defines the agent's role and capabilities, describing what it means to act as an agent in the given environment. It also includes the set of valid operations for agents. The action guidance contains the task goal, detailed semantics of each action, and illustrative usage examples.

---

**System Prompt for Instruction-Tuned Model**

Now output an action from the action list in the correct JSON format, following the reason why you do that. During thinking, you can explore the current page by executing possible actions and memorize the pages you have explored.
The reasoning process and the action are enclosed within `<think> </think>` and `<action> </action>` tags respectively, while the exploration and memory are enclosed within `<explore> </explore>` and `<memory> </memory>` tags respectively.

**Example:**

```
<think> This is the reasoning process. </think>
<explore> Possible actions to explore here </explore>
<memory> Memory for previous pages here </memory>
<action> The final action is \boxed{action here} </action>
```

In the last part of the action, the final exact action must be enclosed within `\boxed{}` with the action format.

---

*Figure 5.* System prompt template specifying the reasoning, exploration, memory, and action generation protocol.

---

**Basic Prompt for Instruction-Tuned Model**

You are an agent who can operate an Android phone on behalf of a user. Based on the user's goal or request, you may:
- Answer back if the request is a question or a chat message (e.g., "What is my schedule for today?").

- Complete tasks described in the request by performing actions step by step on the phone.

When given a user request, you will attempt to complete it step by step. At each step, you are provided with the current screenshot (including the original screenshot and the annotated version with bounding boxes and numeric indexes) and a textual history of previously executed actions. Based on this information and the goal, you must output exactly one action in the correct JSON format.

**Termination Conditions:**
- If the task is completed, output: `{{action_type: status, goal_status: complete}}`

- If the task is infeasible, output: `{{action_type: status, goal_status: infeasible}}`

- If the request is a question, output: `{{action_type: answer, text: <answer_text>}}`

**Action Space:** `{{action_type: click, index: <target_index>}}`
`{{action_type: long_press, index: <target_index>}}`
`{{action_type: input_text, text: <text_input>, index: <target_index>}}`
`{{action_type: keyboard_enter}}`
`{{action_type: navigate_home}}`
`{{action_type: navigate_back}}`
`{{action_type: scroll, direction: <up, down, left, right>, index: <optional_target_index>}}`
`{{action_type: open_app, app_name: <name>}}`
`{{action_type: wait}}`

---

*Figure 6.* Basic prompt defining the agent role, task completion protocol, and available action space.

**Action Guidance for Agentic Decision Making**

This section provides detailed guidance on task completion and action execution. The action guidance specifies the task objective, the semantics of each available action, and concrete usage examples to ensure consistent and executable outputs from the agentic model.

**Task Objective:**
Given a user request and the current environment observation, the agent must select exactly one action at each step to make progress toward completing the task. The agent should reason about the goal, current UI state, and historical actions before choosing the next action.

**General Action Constraints:**
- At each step, output exactly one action in valid JSON format.
- The selected action must be executable under the current UI state.
- Do not hallucinate UI elements or indexes that are not present.
- If the task has been completed, terminate using a `status` action.

**Action Definitions and Examples:**
- `click`: Tap a UI element specified by its index.

      {{action_type: click, index: 5}}

- `long_press`: Perform a long press on a UI element.

      {{action_type: long_press, index: 3}}

- `input_text`: Input text into a text field.

      {{action_type: input_text, text: meeting notes, index: 2}}

- `keyboard_enter`: Press the enter key on the virtual keyboard.

      {{action_type: keyboard_enter}}

- `scroll`: Scroll the screen in a specified direction.

      {{action_type: scroll, direction: down}}

- `navigate_back`: Navigate to the previous screen.

      {{action_type: navigate_back}}

- `navigate_home`: Return to the home screen.

      {{action_type: navigate_home}}

- `open_app`: Open an application by name.

      {{action_type: open_app, app_name: Calendar}}

- `wait`: Take no action and wait for the UI to update.

      {{action_type: wait}}

- `status`: Indicate task completion or infeasibility.

      {{action_type: status, goal_status: complete}}

**Prompt Composition:**
The system prompt is provided to the model as a system-level instruction. The basic prompt and the action guidance are concatenated and supplied as the user prompt, together with task-specific observations, to guide step-by-step decision making.

*Figure 7.* Action guidance specifying task objectives, action semantics, and usage examples for agentic models.

# D. Additional Results

## D.1. Comparison with More Models

To further investigate the performance of EAPO, we compare it with more current strong base or agentic models. As shown in Table 4, a 2B-scale model trained with EAPO significantly outperforms a wide range of strong general LLMs with substantially larger parameter sizes (especially in challenging long-horizon GUI tasks). These results demonstrate that EAPO enables the agent to autonomously explore at appropriate stages of interaction, thereby improving its understanding of the environment during execution rather than relying solely on model scale.

*Table 4.* Success rate using different agentic models on text-based environments. We exhibit the performance advantage with the best baseline and highlight the  best  result.

| Model | ALFworld | WebShop | AndroidWorld | OSWorld |
|---|---|---|---|---|
| *Closed-source Models* | | | | |
| OpenAI CUA o3 (OpenAI, 2025) | 42.31 | 38.73 | 55.43 | 23.00 |
| TianXi-Action-7B (Tian et al., 2024) | 36.57 | 32.46 | 48.93 | 29.81 |
| OpenAI CUA (OpenAI, 2025) | 44.88 | 40.29 | 57.63 | 31.30 |
| Claude-3.7-Sonnet (Anthropic, 2025a) | 48.61 | 44.11 | 60.83 | 35.83 |
| DeepMiner-Mano-7B (Fu et al., 2025) | 46.27 | 41.94 | 62.30 | 40.16 |
| DeepMiner-Mano-72B (Fu et al., 2025) | 54.12 | 49.71 | 68.48 | 53.88 |
| Seed1.5-VL-250717 (Guo et al., 2025a) | 47.53 | 43.09 | 61.75 | 40.18 |
| UI-TARS-2-2509 (Wang et al., 2025a) | 51.21 | 46.85 | 73.38 | 53.11 |
| Claude-4-Sonnet-0929 (Anthropic, 2025b) | 56.09 | 50.97 | 71.60 | 62.88 |
| *Open-source Models* | | | | |
| Qwen3-VL-32B (Bai et al., 2025) | 49.69 | 44.29 | 63.70 | 41.00 |
| Qwen3-VL-235B-A22B (Bai et al., 2025) | 50.81 | 45.05 | 62.00 | 38.10 |
| ZeroGUI (Yang et al., 2025b) | 35.76 | 31.29 | 47.52 | 20.20 |
| UI-TARS-72B-dpo (Qin et al., 2025) | 44.36 | 39.61 | 46.65 | 26.84 |
| UI-TARS-7B (Qin et al., 2025) | 31.82 | 28.55 | 33.04 | 27.52 |
| OpenCUA-7B (Wang et al., 2025c) | 34.69 | 30.70 | 45.10 | 28.20 |
| OpenCUA-32B (Wang et al., 2025c) | 39.99 | 35.48 | 51.66 | 34.79 |
| ARPO (Lu et al., 2025) | 37.20 | 33.17 | 49.31 | 29.90 |
| GUI-Owl-7B (Ye et al., 2025) | 38.47 | 34.06 | 52.04 | 34.79 |
| DART-GUI-7B (Li et al., 2025) | 45.78 | 40.83 | 57.99 | 42.13 |
| *Ours* | | | | |
| EAPO-1.7B/2B | 58.50 ↑2.41 | 53.28 ↑2.31 | 76.36 ↑1.98 | 50.34 ↓12.54 |
| EAPO-4B | 69.00 ↑9.91 | 60.84 ↑9.87 | 79.59 ↑6.21 | 57.89 ↑4.99 |
| EAPO-8B | 76.02 ↑19.93 | 65.58 ↑14.61 | 82.05 ↑8.67 | 64.29 ↑1.41 |

### D.2. Comparison with Training Methods

We evaluate EAPO's performance with varying the model size, including 1.7B, 4B, 8B for text-based environments and 2B, 4B. 8B for GUI-based environments. The comparative results and learning curves are shown in Table 5 and Figs. 8 and 9.

**Summary of key findings.** The results show that EAPO significantly improves agent performance, consistently surpassing existing methods by $20\% - 60\%$, particularly in complex tasks like GUI control. This highlights its ability to obtain dynamic information via adaptive exploration. In addition, EAPO demonstrates faster and more stable convergence, achieving strong performance in fewer training iterations compared to baseline methods, which indicates that learning when to explore effectively reduces unnecessary trial-and-error and accelerates policy optimization.

*Table 5.* Success rate using different model on different environments. We exhibit the performance advantage with the best baseline and highlight the best result.

| Model | Method | ALFworld | WebShop | AndroidWorld | OSWorld |
|---|---|---|---|---|---|
| Qwen-1.7B Qwen-VL-2B | 0-shot | 10.45 | 11.63 | 46.19 | 19.08 |
| | Min-p | 15.34 | 22.48 | 50.86 | 23.96 |
| | OverRIDE | 27.15 | 32.79 | 53.44 | 26.54 |
| | GRPO | 32.84 | 30.22 | 52.38 | 22.87 |
| | DAPO | 40.46 | 33.70 | 56.82 | 25.18 |
| | GiGPO | 42.62 | 35.30 | 56.01 | 28.71 |
| | LAMER | 45.47 | 37.92 | 61.26 | 33.76 |
| | EAPO | 58.50$_{\uparrow 13.1}$ | 53.28$_{\uparrow 15.3}$ | 76.36$_{\uparrow 15.1}$ | 50.34$_{\uparrow 16.6}$ |
| Qwen-4B Qwen-VL-4B | 0-shot | 20.91 | 22.31 | 52.01 | 31.41 |
| | Min-p | 27.53 | 28.09 | 56.60 | 36.23 |
| | OverRIDE | 29.53 | 28.80 | 58.99 | 38.53 |
| | GRPO | 45.23 | 36.14 | 56.13 | 38.65 |
| | DAPO | 48.78 | 38.29 | 60.66 | 43.16 |
| | GiGPO | 51.35 | 40.45 | 59.78 | 47.19 |
| | LAMER | 54.60 | 43.8 | 61.54 | 49.24 |
| | EAPO | 69.00$_{\uparrow 14.4}$ | 60.84$_{\uparrow 17.0}$ | 79.57$_{\uparrow 18.0}$ | 57.89$_{\uparrow 8.6}$ |
| Qwen-8B Qwen-VL-8B | 0-shot | 22.80 | 24.17 | 50.07 | 33.96 |
| | Min-p | 28.76 | 30.34 | 54.71 | 37.90 |
| | OverRIDE | 31.41 | 33.21 | 60.71 | 40.49 |
| | GRPO | 46.13 | 38.57 | 55.48 | 40.36 |
| | DAPO | 51.86 | 40.90 | 61.76 | 47.90 |
| | GiGPO | 56.76 | 42.24 | 58.87 | 50.88 |
| | LAMER | 61.26 | 47.74 | 60.98 | 55.60 |
| | EAPO | 76.02$_{\uparrow 14.8}$ | 65.58$_{\uparrow 17.8}$ | 82.05$_{\uparrow 21.1}$ | 64.29$_{\uparrow 8.6}$ |

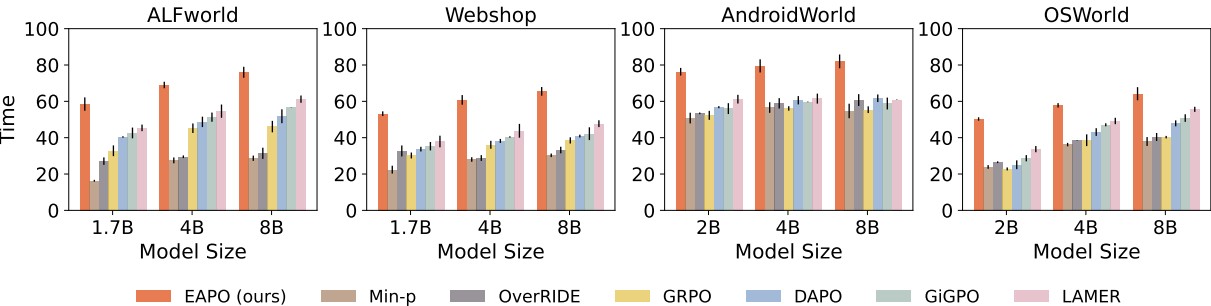

*Figure 8.* Performance with varying model size. Uncertainty intervals depict standard deviation over three seeds. EAPO exhibits higher performance, demonstrating the effectiveness of exploration during test time and its great efficiency in encouraging agents to explore compared to existing methods.

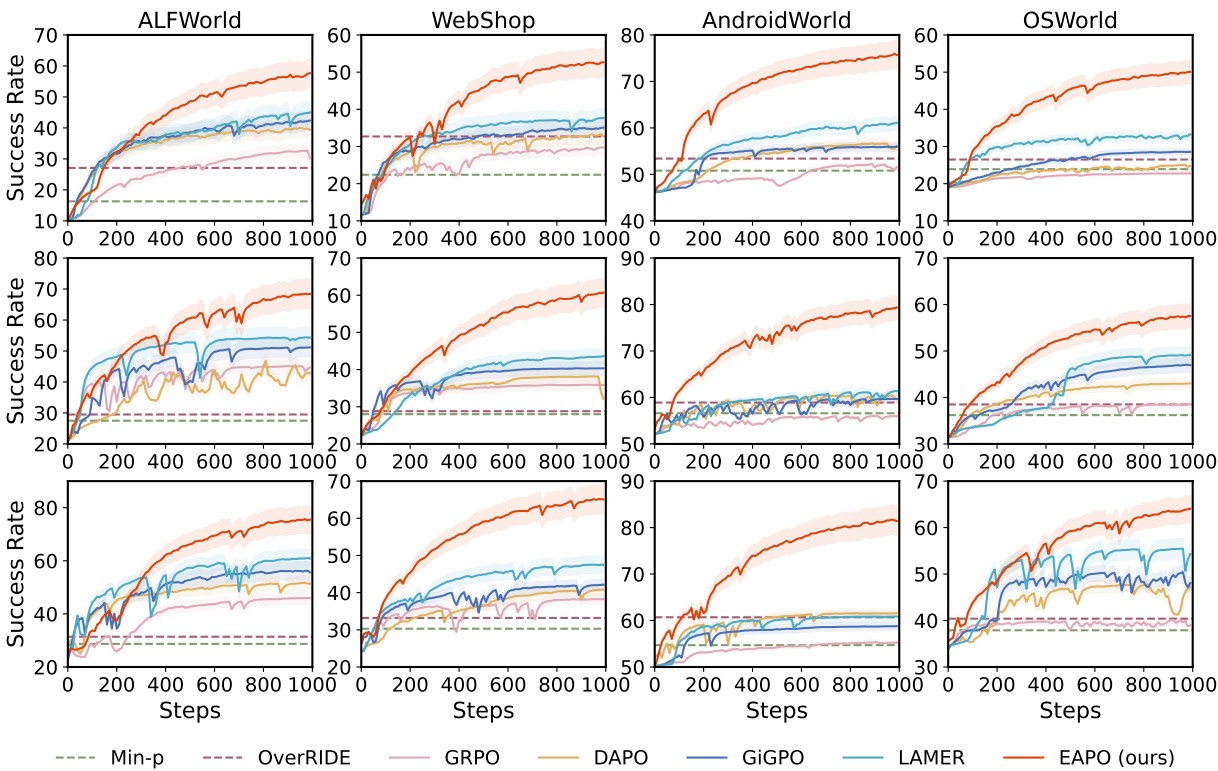

*Figure 9.* Training convergence with varying model size. Uncertainty intervals depict standard deviation over three seeds. EAPO consistently and significantly surpasses existing methods in terms of convergence speed and stability.

## D.3. Generalization

To verify the exploration capability, we demonstrate the performance of applying trained models to unseen scenarios. To be specific, we apply the model trained on AndroidWorld to OSWorld and demonstrated the success rate in each task domain. As shown in Table 6, models trained on AndroidWorld by EAPO consistently achieve strong performance across diverse task domains. Compared to models trained directly on OSWorld, we observe only a slight performance degradation when transferring to OSWorld, indicating that the learned behaviors generalize well across platforms. This result suggests that EAPO primarily captures domain-invariant exploration strategies rather than overfitting to environment-specific interfaces or applications. Consequently, the agent is able to effectively reuse its exploration and decision-making patterns in previously unseen environments, demonstrating robust cross-domain generalization capability.

*Table 6.* Task success rate across different applications. EAPO (AndroidWorld) refers to model trained on AndroidWorld. We highlight the best result.

| Model | *chrome* | *gimp* | *calc* | *impress* | *writer* | *multi_apps* | *os* | *thunderbird* | *vlc* | *vs_code* | **Overall** |
|---|---|---|---|---|---|---|---|---|---|---|---|
| **Closed-source Model** | | | | | | | | | | | |
| OpenAI CUA o3 | 13.04 | 38.46 | 10.64 | 10.64 | 30.43 | 16.53 | 62.50 | 26.67 | 39.18 | 39.13 | 23.00 |
| TianXi-Action-7B | 36.83 | 55.77 | 6.38 | 38.24 | 54.35 | 6.60 | 38.22 | 43.33 | 31.85 | 67.39 | 29.81 |
| OpenAI CUA | 36.87 | 34.62 | 14.89 | 29.70 | 26.09 | 15.81 | 70.83 | 66.67 | 11.76 | 69.57 | 31.30 |
| Claude-3.7-Sonnet | 52.09 | 38.46 | 31.91 | 36.09 | 43.48 | 17.66 | 50.00 | 53.33 | 23.53 | 56.52 | 35.83 |
| DeepMiner-Mano-7B | 39.13 | 69.23 | 27.66 | 42.47 | 56.52 | 17.20 | 50.00 | 73.33 | 35.29 | 78.26 | 40.16 |
| Seed1.5-VL-2507 | 56.52 | 50.00 | 34.78 | 48.91 | 56.52 | 15.35 | 39.13 | 73.33 | 35.29 | 56.52 | 40.18 |
| UI-TARS-2507s | 56.43 | 50.00 | 40.43 | 55.30 | 60.87 | 14.66 | 41.67 | 66.67 | 44.00 | 52.17 | 41.84 |
| Claude-4-Sonnet | 54.26 | 50.00 | 31.91 | 46.72 | 60.87 | 28.49 | 45.83 | 73.33 | 41.18 | 60.87 | 43.88 |
| **Open-source Model** | | | | | | | | | | | |
| Qwen2.5-VL-32B | 8.70 | 3.85 | 0.00 | 0.00 | 8.70 | 2.15 | 8.33 | 6.67 | 0.00 | 8.70 | 3.88 |
| Qwen2.5-VL-72B | 4.35 | 0.00 | 6.38 | 0.00 | 8.70 | 3.23 | 16.67 | 13.33 | 5.88 | 4.35 | 4.99 |
| ZeroGUI | – | – | – | – | – | – | – | – | – | – | 20.20 |
| UI-TARS-72B-dpo | 33.24 | 61.54 | 12.77 | 25.45 | 43.48 | 6.71 | 33.33 | 33.33 | 23.53 | 47.83 | 25.88 |
| UI-TARS-72B-dpo | 32.61 | 73.08 | 6.38 | 23.81 | 34.78 | 8.29 | 37.50 | 60.00 | 17.65 | 52.17 | 26.84 |
| UI-TARS-1.5-7B | 38.34 | 51.92 | 9.57 | 38.21 | 39.13 | 8.94 | 31.25 | 40.00 | 22.44 | 47.83 | 27.52 |
| OpenCUA-7B | 38.61 | 43.59 | 13.22 | 32.60 | 33.33 | 12.11 | 43.47 | 42.22 | 28.31 | 47.10 | 28.20 |
| OpenCUA-32B | 39.77 | 66.67 | 18.44 | 37.60 | 36.23 | 16.21 | 55.07 | 46.67 | 33.33 | 63.31 | 34.79 |
| ARPO | – | – | – | – | – | – | – | – | – | – | 29.90 |
| GUI-Owl-7B | 41.22 | 65.38 | 17.02 | 19.06 | 52.17 | 9.68 | 50.00 | 66.67 | 29.41 | 65.22 | 32.11 |
| DART-GUI-7B | 52.09 | 76.92 | 19.15 | 48.80 | 60.86 | 16.69 | 62.50 | 60.00 | 39.30 | 69.57 | 42.13 |
| **ours** | | | | | | | | | | | |
| EAPO-2B (AndroidWorld) | 44.08 | 49.78 | 49.38 | 43.37 | 41.15 | 40.64 | 48.88 | 43.08 | 45.29 | 56.57 | 48.83 |
| EAPO-4B (AndroidWorld) | 50.96 | 57.48 | 55.92 | 50.11 | 47.56 | 46.97 | 56.47 | 50.21 | 52.34 | 65.30 | 55.28 |
| EAPO-8B (AndroidWorld) | 56.85 | 63.39 | 64.09 | 55.88 | 53.00 | 52.34 | 62.97 | 56.85 | 58.44 | 72.58 | 60.64 |

## D.4. Impact of Discount $\gamma$

To assess the effect of the discount factor, we carry out experiments by varying the discount factor $\gamma$ from $0.5$ to $1.0$. As illustrated in Fig. 10, a larger value of $\gamma$ results in a higher encouragement on exploration, leading agents to obtain adequate information before making the final decision. However, excessively large $\gamma$ may induce over-exploration, causing the agent to accumulate redundant or noisy information, which can interfere with action generation and hinder effective decision-making. Conversely, a smaller value of $\gamma$ substantially attenuates the rewards obtained through exploration, causing the optimization process to degenerate into a conventional goal-oriented GRPO. Therefore, it is crucial to appropriately adjust $\gamma$ to control the degree of exploration, thereby achieving a better balance between information gathering and task execution performance.

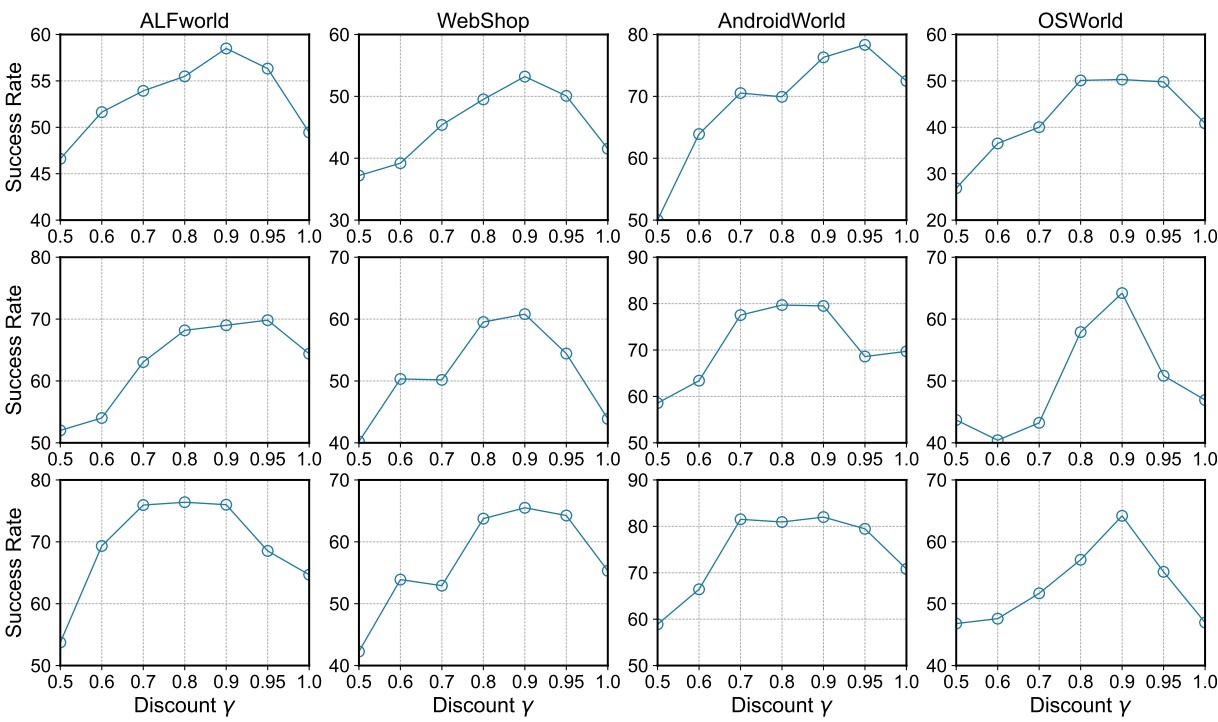

*Figure 10.* Performance with varying discount $\gamma$ when training Qwen models in text-based environments and Qwen-VL models in GUI-based environments.

We further conduct experiment to verify the average steps when varying $\gamma$. As illustrated in Section D.4, the average number of steps consistently grows across all environments with increasing discount $\gamma$. This trend indicates that a larger $\gamma$ encourages the agent to promote more extensive exploration before committing to final decisions. In particular, when $\gamma$ approaches $1.0$, the agent tends to over-explore, resulting in significantly longer trajectories.

Considering both inference efficiency and task performance, we set $\gamma = 0.9$ across all environment, which provides a better trade-off, enabling sufficient information gathering while avoiding excessive and redundant exploration.

| Discount $\gamma$ | 0.5 | 0.6 | 0.7 | 0.8 | 0.9 | 0.95 | 1.0 |
|---|---|---|---|---|---|---|---|
| **ALFWorld** | 12.4 | 14.8 | 17.3 | 19.6 | 22.5 | 31.8 | 43.9 |
| **WebShop** | 10.1 | 12.7 | 15.4 | 17.9 | 19.8 | 27.6 | 38.7 |
| **AndroidWorld** | 13.2 | 16.5 | 19.8 | 21.6 | 22.7 | 30.9 | 44.8 |
| **OSWorld** | 11.8 | 15.1 | 18.3 | 20.1 | 21.3 | 29.4 | 42.6 |

*Table 7.* Comparison of average steps.

## D.5. Impact of Sampling Group Size $G$

To assess the effect of the group size for sampling, we carry out experiments by varying the group size $G$ from 4 to 32. As illustrated in Fig. 11, increasing the group size generally leads to more stable policy optimization, as a larger set of sampled trajectories provides a better estimate of relative advantages within each group. However, the performance gain gradually saturates as $G$ becomes large, since excessively increasing the group size mainly introduces additional computational overhead without bringing proportional improvement. This suggests that a moderate group size is sufficient to balance optimization stability and computational efficiency in EAPO.

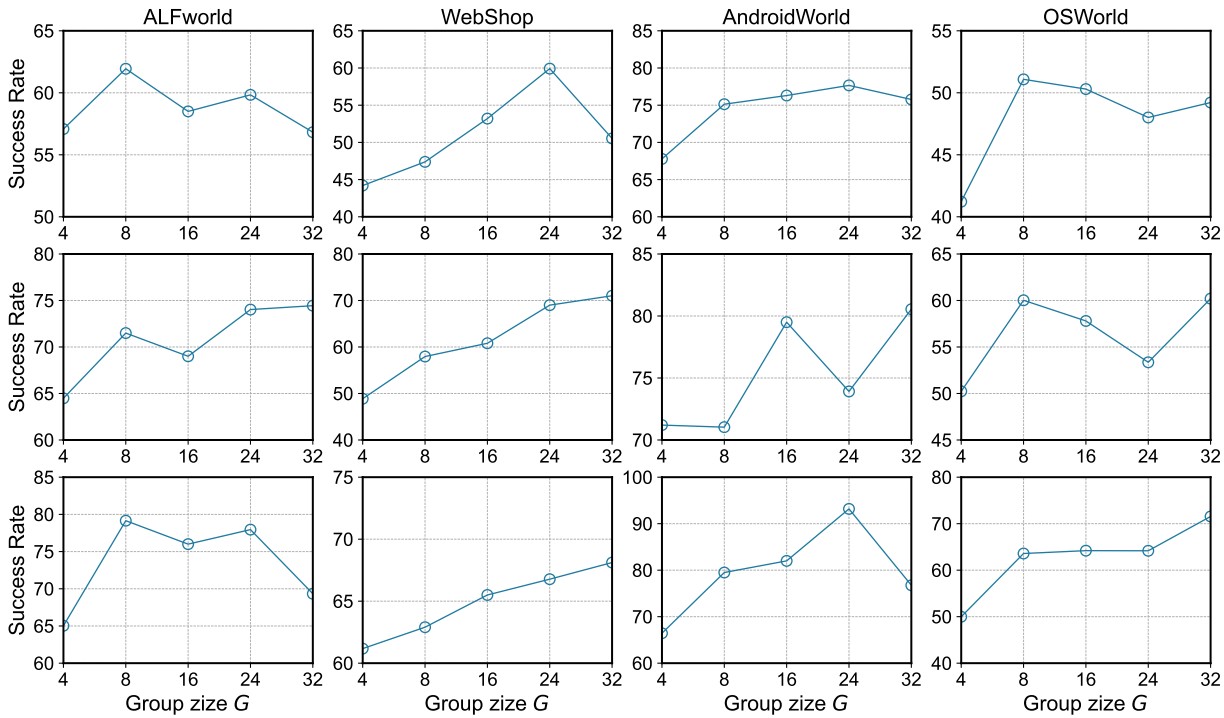

*Figure 11.* Performance with varying group size $G$ when training Qwen models in text-based environments and Qwen-VL models in GUI-based environments.

### D.6. Impact of KL loss coefficency $\lambda$

To assess the effect of the KL coefficient, we carry out experiments by varying the coefficient $\lambda$ from $0.005$ to $1$. As illustrated in Fig. 12, the performance first improves and then degrades as $\lambda$ increases. When $\lambda$ is small, the KL regularization is weak, allowing the policy to deviate aggressively from the reference model, which may lead to unstable updates and suboptimal optimization. Increasing $\lambda$ introduces a stronger constraint that stabilizes training and helps preserve useful prior knowledge, resulting in improved performance. However, when $\lambda$ becomes overly large, the policy is excessively restricted to remain close to the reference model, which suppresses effective policy improvement and limits exploration, ultimately causing performance degradation.

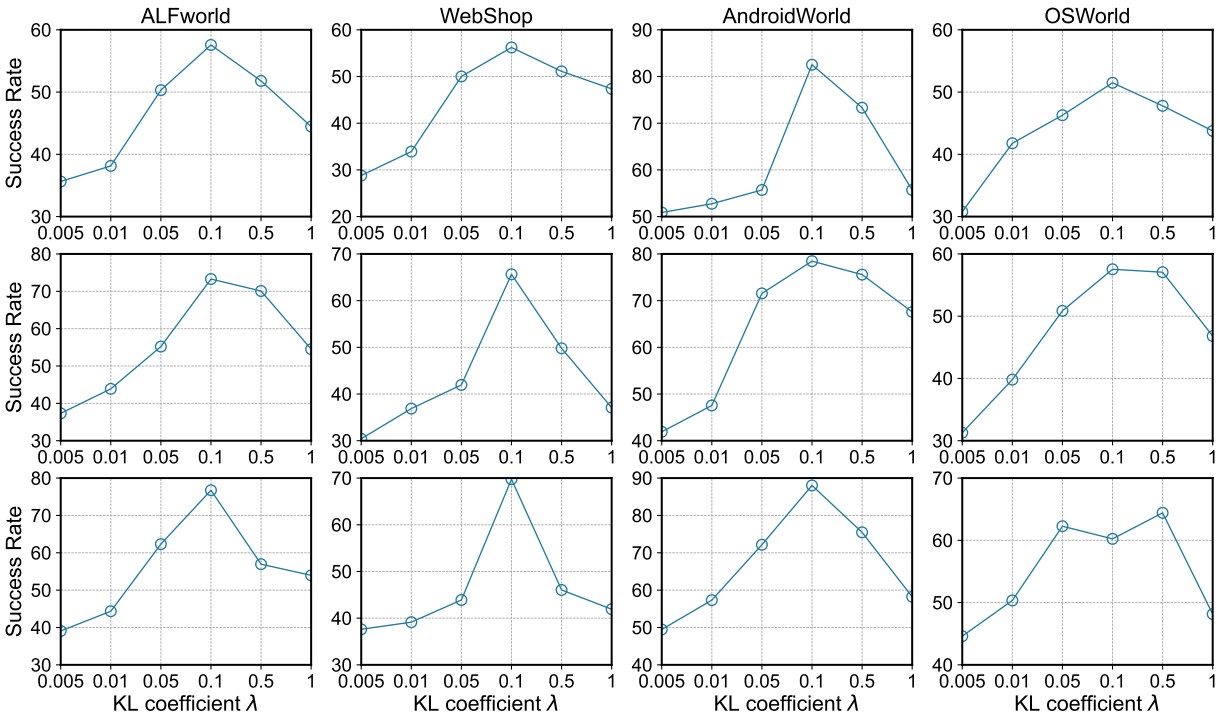

*Figure 12.* Performance with varying KL coefficient $\lambda$ when training Qwen models in text-based environments and Qwen-VL models in GUI-based environments.

## D.7. Ablation Studies and Complementary Experiments

### D.7.1. CONVERGENCE OF MEMORY DISTRIBUTION

We verify the convergence of solving Problem (8) by displaying the value of loss. As shown in Fig. 13, it works well in all environments and often converges in 600 to 800 training steps.

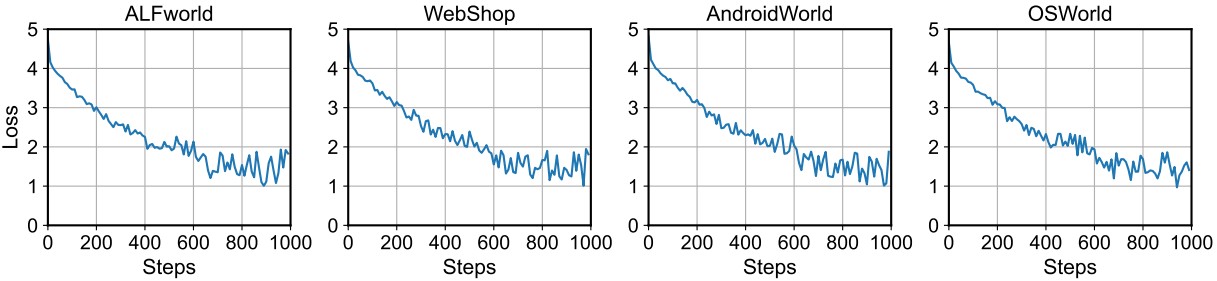

*Figure 13.* The value of reward loss when training Qwen-1.7B in text-based environments and Qwen-VL-2B in GUI-based environments.

### D.7.2. CONVERGENCE OF POLICY MODEL

We verify the convergence of policy updating by displaying the success rate by training steps. As shown in Fig. 13, it works well in all environments and often converges in 600 to 800 training steps.

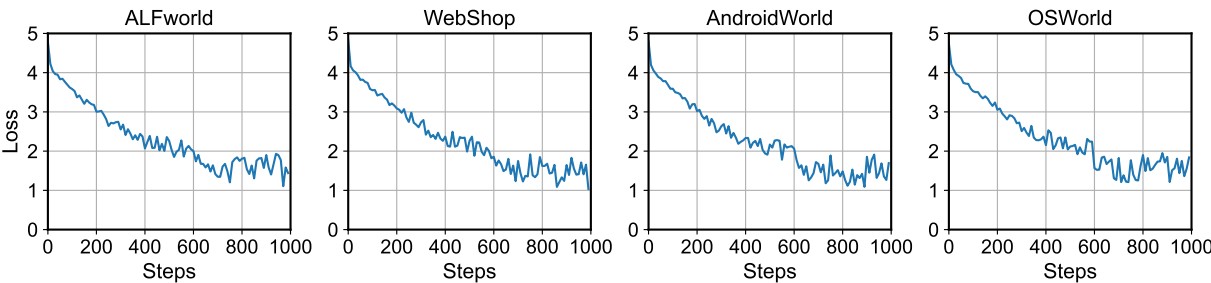

*Figure 14.* The value of policy loss when training Qwen-1.7B in text-based environments and Qwen-VL-2B in GUI-based environments.

### D.7.3. REWARD

We demonstrate how each part of the reward model (format, exploratory, and task success) changes during training in Fig. 15. The results exhibits consistent policy improvement, indicating the efficiency of our reward model and policy optimization method. Of note, our model quickly learns the format requirements in less than rounds, indicating that the proposed reward will not shift learning toward parseable output rather than doing high-value exploration.

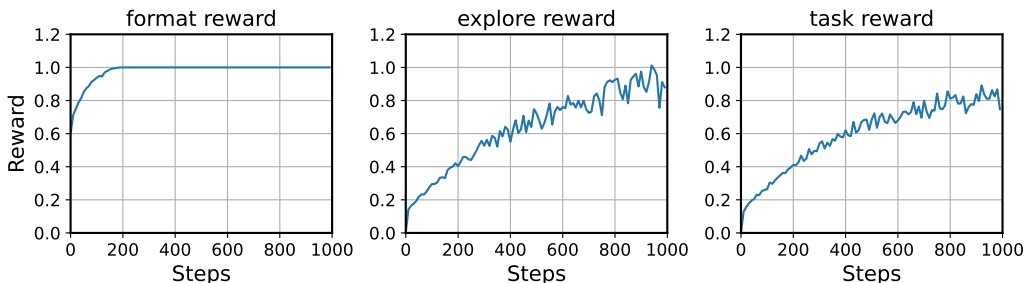

*Figure 15.* The value of each part of the reward model when training Qwen-VL-2B in AndroidWorld.

### D.7.4. STEP-LEVEL GROUP SIZE

We examine how the distribution of step-level groups evolves throughout training to better understand the utility of exploration-aware grouping. We use Qwen3-1.7B for text-based environments and Qwen3-VL-2B for GUI-based environments. We track changes in step-level group sizes throughout training.

As illustrated in Fig. 16, at the early stage of training, the proportion of step-level groups with larger sizes increases noticeably as training proceeds, indicating that identical states are visited multiple times more frequently. This phenomenon suggests that the agent gradually learns to actively explore the environment rather than committing to greedy decisions. As training continues, the group-size distribution becomes stable, reflecting that the agent learns to judiciously decide when exploration is necessary, instead of conservatively exploring redundant states. This stabilized behavior indicates a more balanced trade-off between exploration and exploitation.

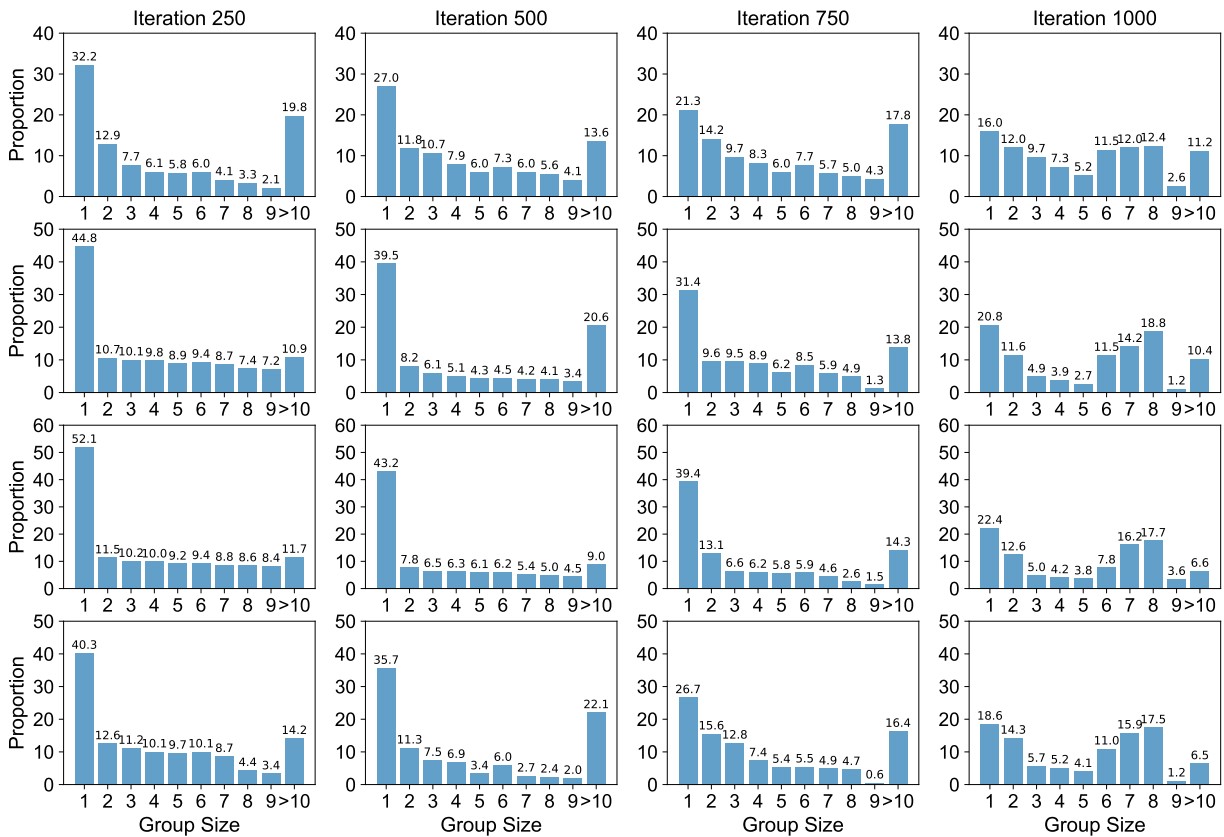

*Figure 16.* Caption

### D.7.5. EXPLORATION DEGREE

To validate that EAPO can help agents to overcome indiscriminate exploration, we demonstrate the exploration degree during training. The exploration degree is defined as the fraction of states that are revisited multiple times among all states, serving as an indicator of the agent's tendency to explore certain states and revisit them to make the final decision.

As shown in Fig. 17, the exploration degree initially increases, indicating that the agent actively explores and revisits informative states to accumulate sufficient contextual evidence. As training proceeds, the exploration degree gradually converges, suggesting that the agent learns to selectively explore only when necessary rather than repeatedly revisiting states in a conservative or indiscriminate manner. This behavior demonstrates that EAPO enables agents to balance exploration and exploitation in a principled way, leading to more efficient decision-making and stable performance improvement.

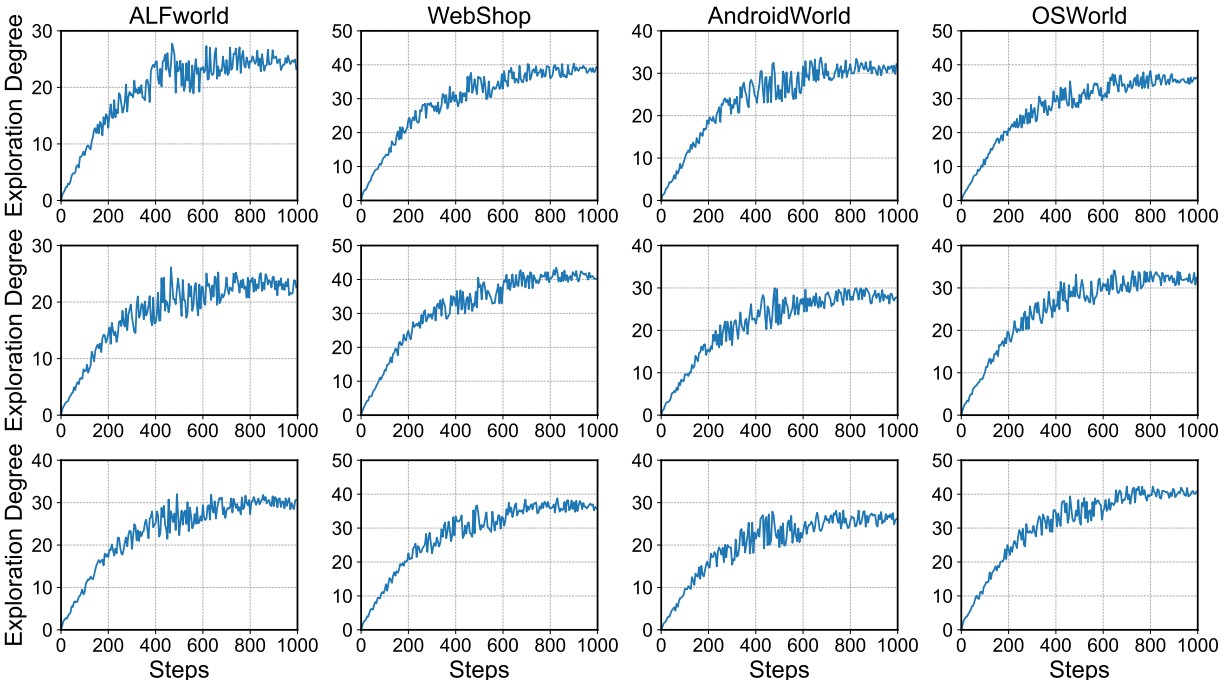

*Figure 17.* Exploration degree with varying model size. EAPO exhibits increasing exploration degree at the beginning as it teaches agents to obtain dynamic information by exploration and converges at a certain level as it balances exploration and exploitation.

### D.7.6. ABLATION

In this section, we assess the effect of key components by ablating them under the same setting.

Without exploratory reward, agents fail to evaluate the usefulness of each exploratory action, causing ineffective exploration and premature convergence to suboptimal behaviors, which leads to significantly degraded performance across all environments. A keen reader may notice that EAPO loses its competitiveness without SFT. We clarify that exploration accuracy and rollback capability are inherently coupled. Even if the agent learns to explore effectively, the acquired information cannot be properly utilized without the ability to rollback to previous decision points. As a result, removing the SFT stage—which provides initial rollback capability—leads to a noticeable performance drop. This does not imply that SFT is responsible for the final performance, but rather that it serves as an enabling component for effective exploration-aware learning.

Without exploration-aware grouping, exploratory actions and task-execution actions are mixed within the same optimization groups. As training progresses, exploration actions are increasingly underestimated, causing both the exploration degree and task performance to first increase and then collapse, indicating unstable exploration and impaired long-term optimization.

Without format reward, the agent fails to consistently follow the predefined output structure for exploration signals and memory updates. This prevents the agent from effectively organizing, storing, and reusing information obtained through exploration, thereby limiting its ability to leverage exploratory behaviors to improve subsequent decision-making and task completion.

*Table 8.* Ablation study on exploratory reward, exploration-aware grouping, and format reward.

| Model | Method | ALFworld | WebShop | AndroidWorld | OSWorld |
|---|---|---|---|---|---|
| Qwen-1.7B Qwen-VL-2B | w/o exploratory | $37.6_{\downarrow 20.9}$ | $30.3_{\downarrow 22.9}$ | $51.0_{\downarrow 25.2}$ | $23.7_{\downarrow 26.6}$ |
| | w/o grouping | $38.8_{\downarrow 19.7}$ | $42.0_{\downarrow 11.2}$ | $61.3_{\downarrow 15.0}$ | $25.4_{\downarrow 24.9}$ |
| | w/o format | $40.2_{\downarrow 18.3}$ | $45.8_{\downarrow 7.4}$ | $68.6_{\downarrow 7.7}$ | $33.7_{\downarrow 16.6}$ |
| | EAPO | $58.5_{\uparrow 0.0}$ | $53.2_{\uparrow 0.0}$ | $76.3_{\uparrow 0.0}$ | $50.3_{\uparrow 0.0}$ |
| Qwen-4B Qwen-VL-4B | w/o exploratory | $46.2_{\downarrow 22.8}$ | $40.9_{\downarrow 19.9}$ | $62.5_{\downarrow 17.0}$ | $43.0_{\downarrow 14.8}$ |
| | w/o grouping | $47.5_{\downarrow 21.5}$ | $43.3_{\downarrow 17.5}$ | $66.5_{\downarrow 13.0}$ | $43.3_{\downarrow 14.5}$ |
| | w/o format | $59.8_{\downarrow 9.2}$ | $46.0_{\downarrow 14.8}$ | $70.5_{\downarrow 9.0}$ | $51.2_{\downarrow 6.6}$ |
| | EAPO | $69.0_{\uparrow 0.0}$ | $60.8_{\uparrow 0.0}$ | $79.5_{\uparrow 0.0}$ | $57.8_{\uparrow 0.0}$ |
| Qwen-8B Qwen-VL-8B | w/o exploratory | $48.0_{\downarrow 28.0}$ | $43.3_{\downarrow 22.2}$ | $63.1_{\downarrow 18.9}$ | $48.3_{\downarrow 15.9}$ |
| | w/o grouping | $55.3_{\downarrow 20.7}$ | $44.4_{\downarrow 21.1}$ | $72.5_{\downarrow 9.5}$ | $48.9_{\downarrow 15.3}$ |
| | w/o format | $59.7_{\downarrow 16.3}$ | $50.0_{\downarrow 15.5}$ | $73.6_{\downarrow 8.4}$ | $53.9_{\downarrow 10.3}$ |
| | EAPO | $76.0_{\uparrow 0.0}$ | $65.5_{\uparrow 0.0}$ | $82.0_{\uparrow 0.0}$ | $64.2_{\uparrow 0.0}$ |

To further validate the effect of removing exploration-aware grouping, we demonstrate the convergence speed and exploration degree of ablating exploration-aware grouping in Figs. 18 and 19.

We observe that, after ablating exploration-aware grouping, both the exploration degree and task performance exhibit a rise-then-fall trend during training. Specifically, the initial increase indicates that the agent can still benefit from short-term exploration when grouping is removed. However, as training progresses, exploratory actions and task-completing actions obtained after exploration tend to be grouped together, causing the value of exploratory actions to be underestimated during optimization. This result highlights the importance of exploration-aware grouping in maintaining stable exploration and sustained performance improvement.

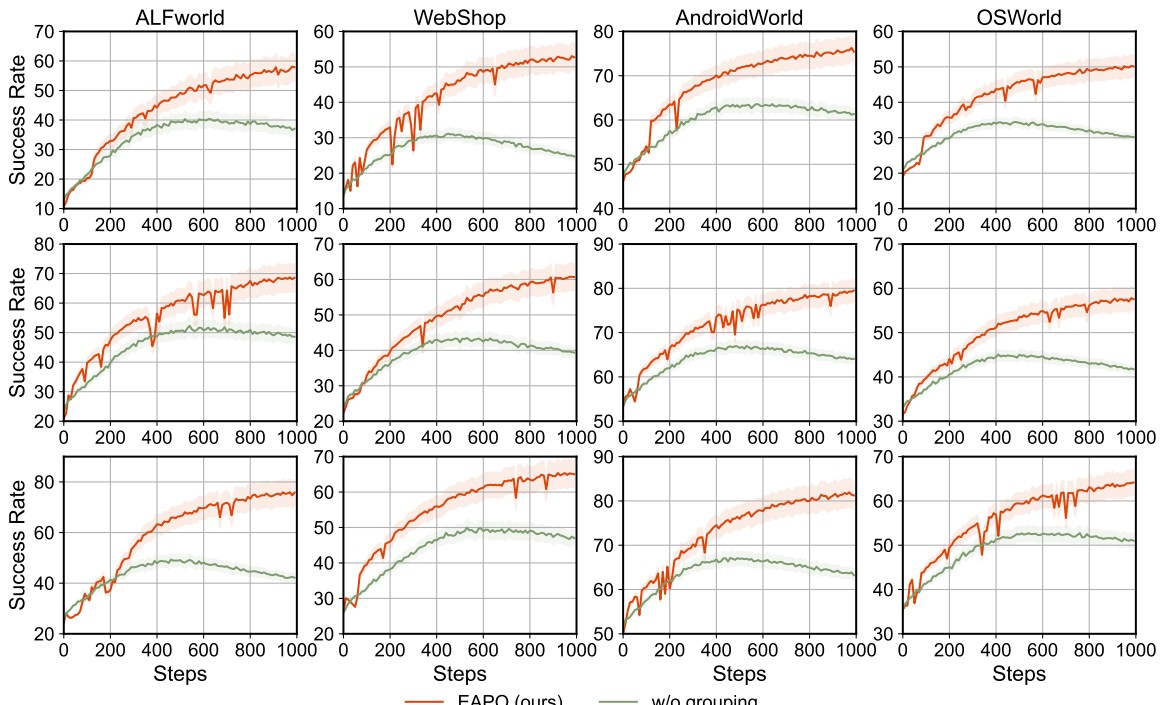

*Figure 18.* Training convergence comparison between EAPO and ablating exploration-aware grouping.

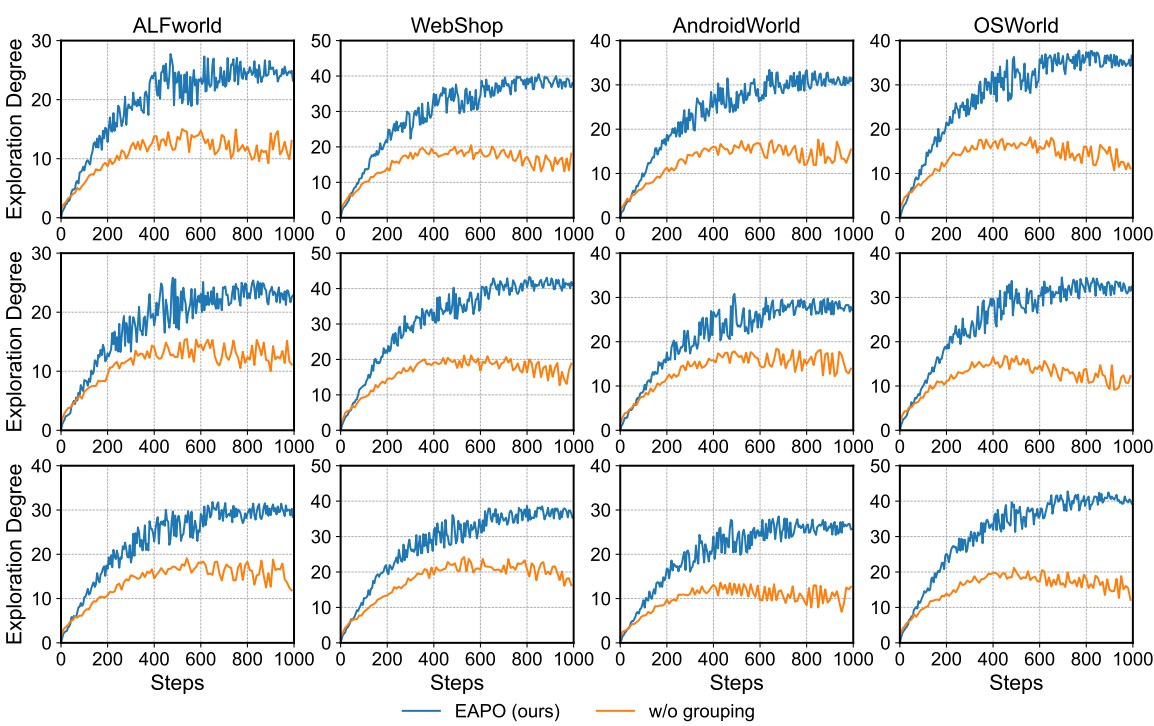

*Figure 19.* Exploration degree comparison between EAPO and ablating exploration-aware grouping.

### D.7.7. RUN-TIME

To demonstrate the training efficiency of our method, we verify its runtime across all the environments. Specifically, we evaluate the runtime of EAPO compared with baseline algorithms utilizing the same model size on 8 NVIDIA H800 GPUs.

We evaluate the total runtime of one step. As illustrated in Fig. 20, the total runtime is approximately twice than other group-based methods, as it involves training an additional reward model. Despite this overhead, the total runtime remains practically manageable and does not hinder scalability.

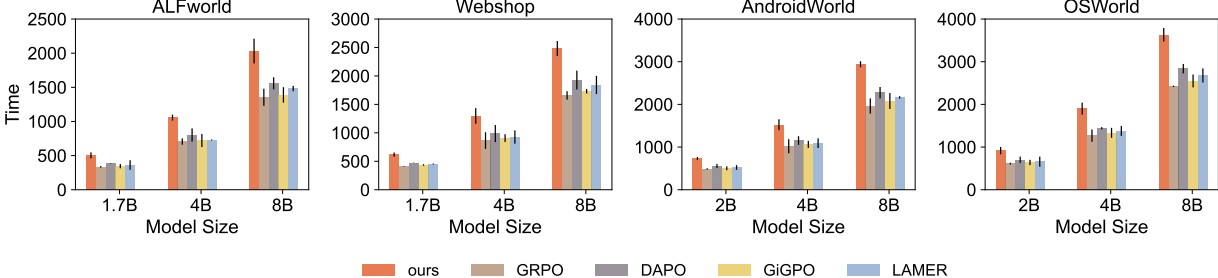

*Figure 20.* Runtime when varying the size of models. Uncertainty intervals depict standard deviation over three seeds.

Further, we demonstrate the time cost of each component. As illustrated by Fig. 21, the GRPO optimization time remains unchanged compared to the baseline, and the additional computational cost of our method mainly comes from training the reward model, grouping the state-action transitions, and inferring the reward for policy advantage estimation.

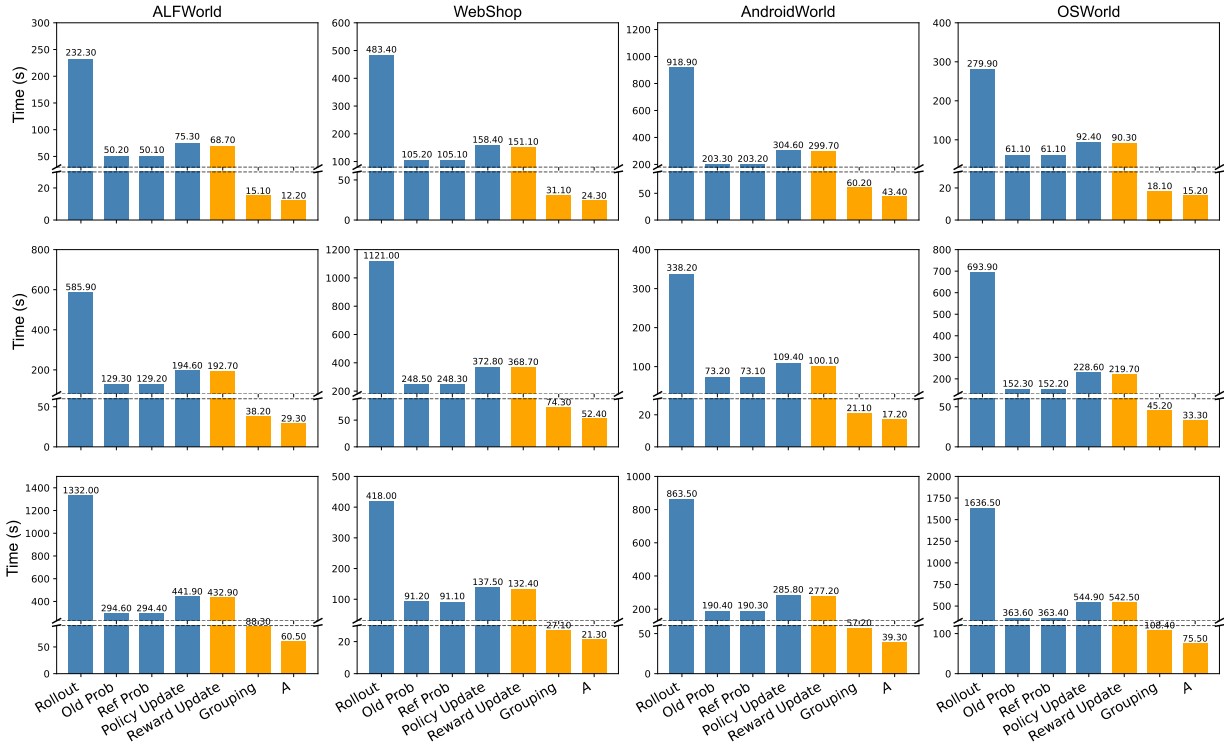

*Figure 21.* Runtime of each components when varying the size of models.

As for inference time, we present comparison of the average step in Section D.7.7 and observe that our method indeed has additional runtime due to exploration. To avoid meaningless exploration, we introduce a discount factor $\gamma$ so that our method maintains only a modest increase (at most 10) in average step. To be specific, we apply a discount to the exploratory gain since the benefit of exploration is not immediate – requiring at least one step to observe a new state and a subsequent step to synthesize the information. It guides the agent to carry out exploration only when the anticipated utility 'outweighs' the latency cost, which will avoid meaningless exploration even when the agent has obtained enough information.

| Method | Min-p | OverRIDE | GRPO | DAPO | GiGPO | LAMER | EAPO (ours) |
|---|---|---|---|---|---|---|---|
| **ALFWorld** | 61.3 | 57.6 | 43.9 | 40.7 | 38.2 | 49.3 | 58.5 |
| **WebShop** | 42.5 | 36.8 | 23.1 | 21.4 | 19.7 | 28.6 | 33.2 |
| **AndroidWorld** | 18.6 | 24.3 | 16.9 | 17.5 | 19.1 | 17.2 | 22.7 |
| **OSWorld** | 21.4 | 27.8 | 19.6 | 20.3 | 22.0 | 20.1 | 21.3 |

*Table 9.* Comparison of average steps.

# E. Case Study

We use a task in OSWorld (Xie et al., 2024). For each step, we present the instructions, screenshots, thoughts, explorations, memories, and actions.

**Summary of key findings.** At Step 5, the agent initiates an exploratory action to gather additional information about the environment. In Step 6, the agent identifies that the explored path leads to an incorrect state, and accordingly performs a rollback to a previous state while incorporating the acquired information. Leveraging this refined understanding, the agent selects the correct action in Step 7, ultimately progressing toward successful task completion. This example illustrates how the agent learns to explore selectively, detect mistakes, and recover through informed backtracking, enabling more robust and effective decision-making in complex environments.

**Step 1**

**System:**

During thinking, you can explore the current page by executing possible actions and memorize the pages you have explored. tags respectively, and the exploration and memory are enclosed within <explore> </explore> and <memory> </memory> tags respectively. For example, <think> This is the reasoning process. </think> <explore> possible actions to explore here </explore> <memory> memory for previous pages here </memory> <think> This is the reasoning process. </think> <action> The final action is \boxed{action here} </action>. In the last part of the action, the final exact action is enclosed within \boxed{} with the action format.

**User:**

You are a GUI agent. You are given a task and your action history, with screenshots. You need to perform the next action to complete the task.

Browse the natural products database.

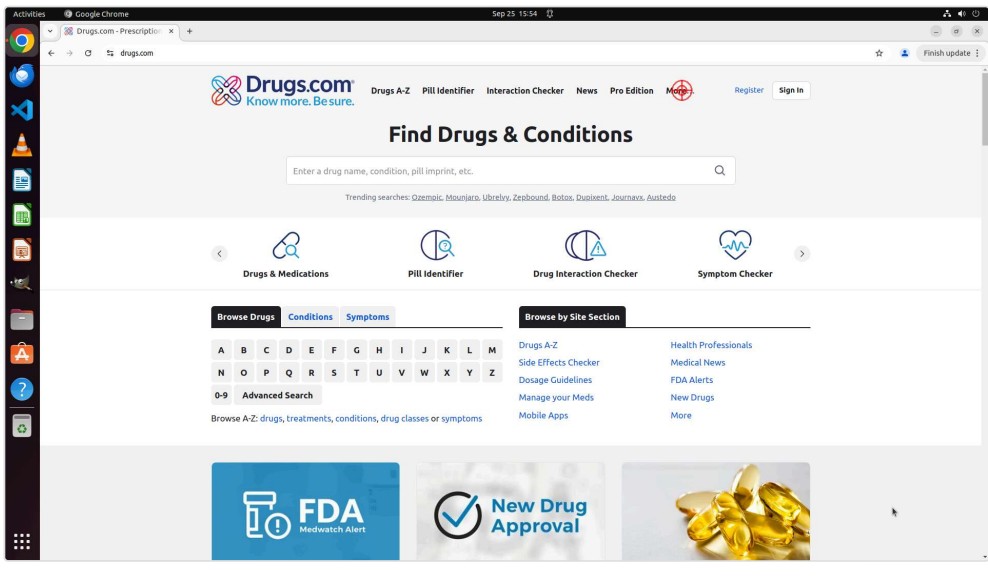

Mouse coordinates: [(1321, 162)] (Action: click)

**Assistant:**

**Thought:** I noticed that there's an option labeled "More..." in the top navigation bar, which seems like it might lead me to additional features.
**Explore:** Since I'm looking for information on natural products, this button should be worth clicking on. Let me go ahead and click it to see what options are available inside.
**Memory:** Nothing.
**Action:** click(start_box='<|box_start|>(1329,164)<|box_end|>')

*Figure 22.* Visualization of EAPO at step 1.

**Step 2**

**System:**

During thinking, you can explore the current page by executing possible actions and memorize the pages you have explored. tags respectively, and the exploration and memory are enclosed within <explore> </explore> and <memory> </memory> tags respectively. For example, <think> This is the reasoning process. </think> <explore> possible actions to explore here </explore> <memory> memory for previous pages here </memory> <think> This is the reasoning process. </think> <action> The final action is \boxed{action here} </action>. In the last part of the action, the final exact action is enclosed within \boxed{} with the action format.

**User:**

You are a GUI agent. You are given a task and your action history, with screenshots. You need to perform the next action to complete the task.

Browse the natural products database.

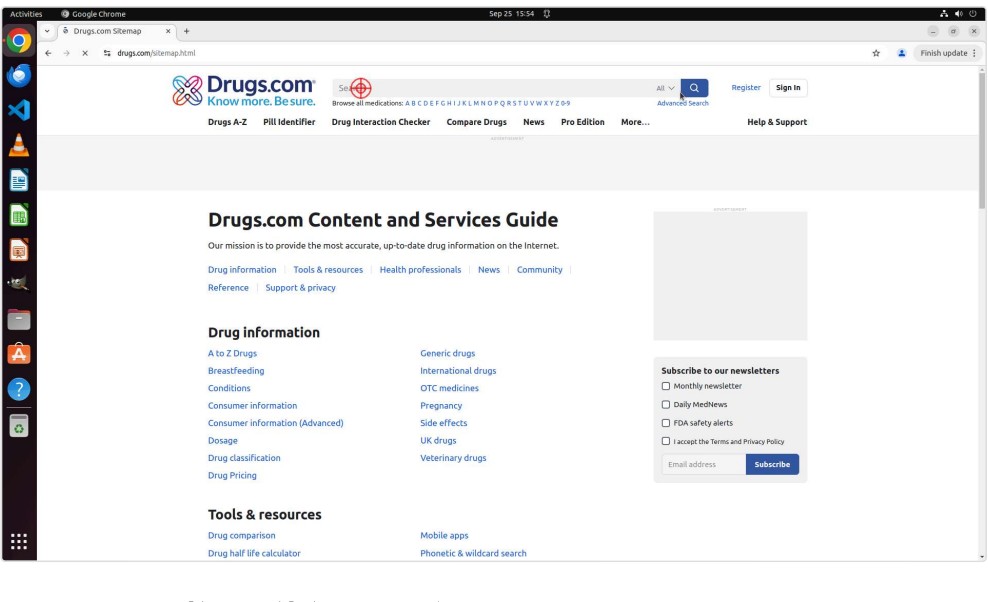

```
Mouse coordinates: [(701, 158)] (Action: click)
```

**Assistant:**

**Thought:** I just clicked on the "More..." button, but it seems like I couldn't find the entry point for natural products. However, that's okay; let me try a different approach.
**Explore:** There's a search box at the top of the page, so searching directly might be quicker. I'll click on this search box first and then type in keywords related to natural products.
**Memory:** At the beginning page, I find a place for searching.
**Action:** click(start_box='<|box_start|>(705,160)<|box_end|>')

*Figure 23.* Visualization of EAPO at step 2.

**Step 3**

**System:**

During thinking, you can explore the current page by executing possible actions and memorize the pages you have explored. tags respectively, and the exploration and memory are enclosed within <explore> </explore> and <memory> </memory> tags respectively. For example, <think> This is the reasoning process. </think> <explore> possible actions to explore here </explore> <memory> memory for previous pages here </memory> <think> This is the reasoning process. </think> <action> The final action is \boxed{action here} </action>. In the last part of the action, the final exact action is enclosed within \boxed{} with the action format.

**User:**

You are a GUI agent. You are given a task and your action history, with screenshots. You need to perform the next action to complete the task.

Browse the natural products database.

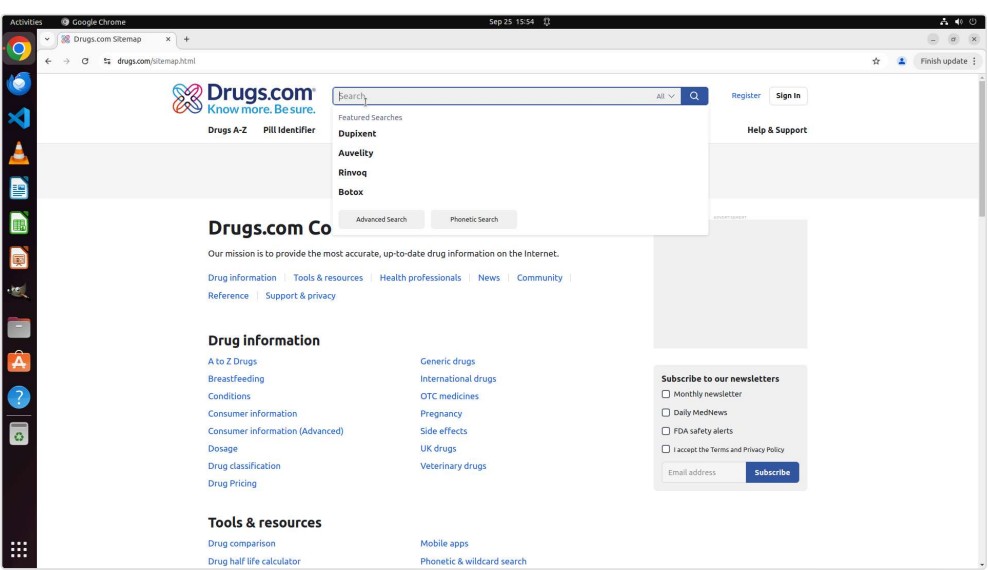

**Assistant:**

**Thought:** The search box is now active, and I'm going to type in the keyword "natural products."
**Explore:** This should help us quickly locate the relevant database content. After all, searching directly is much more efficient than sifting through each category one by one.
**Memory:** I have clicked more at the beginning page and the search box popped out.
**Action:** type(content='natural products')

*Figure 24.* Visualization of EAPO at step 3.

**Step 4**

**System:**

During thinking, you can explore the current page by executing possible actions and memorize the pages you have explored. tags respectively, and the exploration and memory are enclosed within <explore> </explore> and <memory> </memory> tags respectively. For example, <think> This is the reasoning process. </think> <explore> possible actions to explore here </explore> <memory> memory for previous pages here </memory> <think> This is the reasoning process. </think> <action> The final action is \boxed{action here} </action>. In the last part of the action, the final exact action is enclosed within \boxed{} with the action format.

**User:**

You are a GUI agent. You are given a task and your action history, with screenshots. You need to perform the next action to complete the task.

Browse the natural products database.

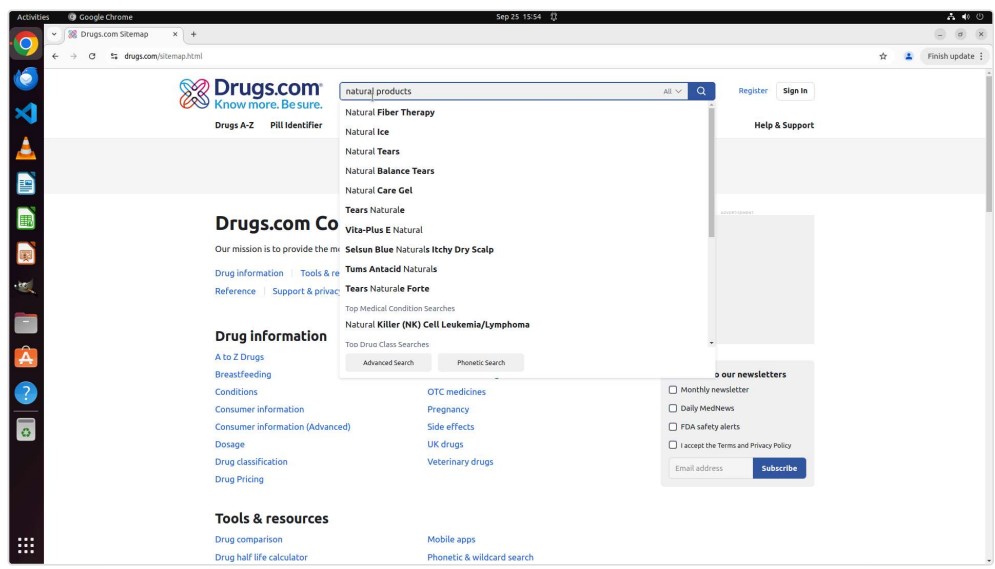

**Assistant:**

**Thought:** Alright, I've finished entering the keywords.
**Explore:** Now, all I need to do is gently press the Enter key, and I'll be able to see detailed information about natural products. Let's start exploring!
**Memory:** I have start the search and enter the keywords for searching.
**Action:** hotkey(key='enter')

*Figure 25.* Visualization of EAPO at step 4.

**Step 5**

**System:**

During thinking, you can explore the current page by executing possible actions and memorize the pages you have explored. tags respectively, and the exploration and memory are enclosed within <explore> </explore> and <memory> </memory> tags respectively. For example, <think> This is the reasoning process. </think> <explore> possible actions to explore here </explore> <memory> memory for previous pages here </memory> <think> This is the reasoning process. </think> <action> The final action is \boxed{action here} </action>. In the last part of the action, the final exact action is enclosed within \boxed{} with the action format.

**User:**

You are a GUI agent. You are given a task and your action history, with screenshots. You need to perform the next action to complete the task.

Browse the natural products database.

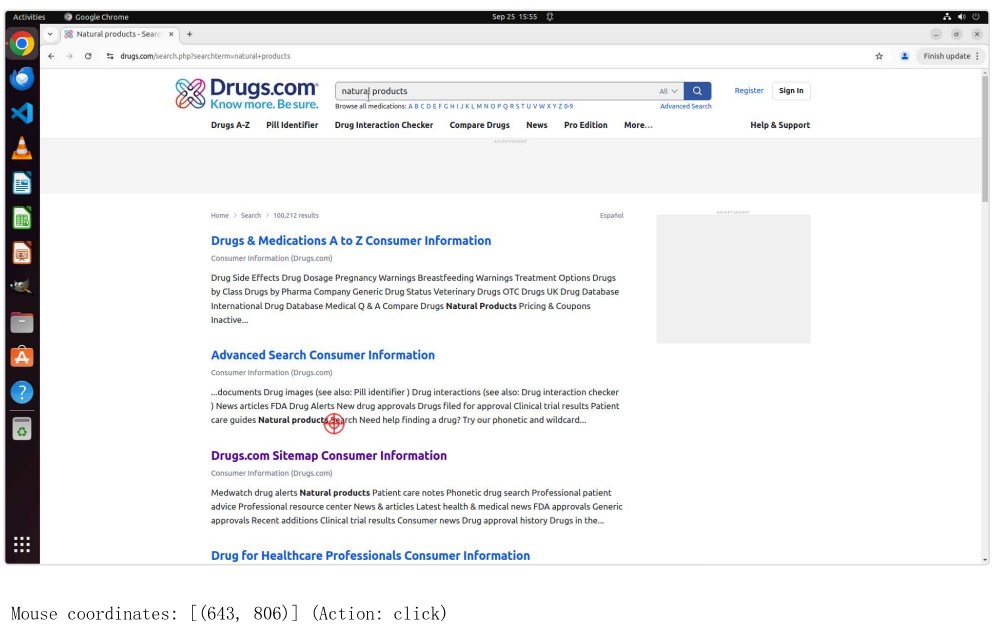

```
Mouse coordinates: [(643, 806)] (Action: click)
```

**Assistant:**
**Thought:** Ah, I found it! In the search results, there's a link labeled "Natural Products," which is exactly what we were looking for.
**Explore:** It seems like this will take us directly to the natural products database page. Let me click on this link to continue exploring.
**Memory:** I have search the "Natural Products," and I should memorize the result after searching in the following steps.
**Action:** click(start_box='<|box_start|>(647,815)<|box_end|>')

*Figure 26.* Visualization of EAPO at step 5. The agent finds multiple possible actions and it decide to explore one by one.

**Step 6**

**System:**

During thinking, you can explore the current page by executing possible actions and memorize the pages you have explored. tags respectively, and the exploration and memory are enclosed within <explore> </explore> and <memory> </memory> tags respectively. For example, <think> This is the reasoning process. </think> <explore> possible actions to explore here </explore> <memory> memory for previous pages here </memory> <think> This is the reasoning process. </think> <action> The final action is \boxed{action here} </action>. In the last part of the action, the final exact action is enclosed within \boxed{} with the action format.

**User:**

You are a GUI agent. You are given a task and your action history, with screenshots. You need to perform the next action to complete the task.

Browse the natural products database.

```
Mouse coordinates: [(89, 88)] (Action: click)
```

**Assistant:**

**Thought:** Oh no, I accidentally clicked into the advanced search interface, which isn't what I was looking for. What I actually need is the natural products database page.
**Explore:** Let me hit the back button in my browser to return to the previous search results page and continue from there.
**Memory:** We can type extra keywords in the advanced search interface but there is no natural products database page.
**Action:** click(start_box='<|box_start|>(90,89)<|box_end|>')

*Figure 27.* Visualization of EAPO at step 6. The agent realizes that it chooses the wrong action, memorze this state as additional information to understand the environment, and perform an action to rollback to the orginal state.

**Step 7**

**System:**

During thinking, you can explore the current page by executing possible actions and memorize the pages you have explored. tags respectively, and the exploration and memory are enclosed within <explore> </explore> and <memory> </memory> tags respectively. For example, <think> This is the reasoning process. </think> <explore> possible actions to explore here </explore> <memory> memory for previous pages here </memory> <think> This is the reasoning process. </think> <action> The final action is \boxed{action here} </action>. In the last part of the action, the final exact action is enclosed within \boxed{} with the action format.

**User:**

You are a GUI agent. You are given a task and your action history, with screenshots. You need to perform the next action to complete the task.

Browse the natural products database.

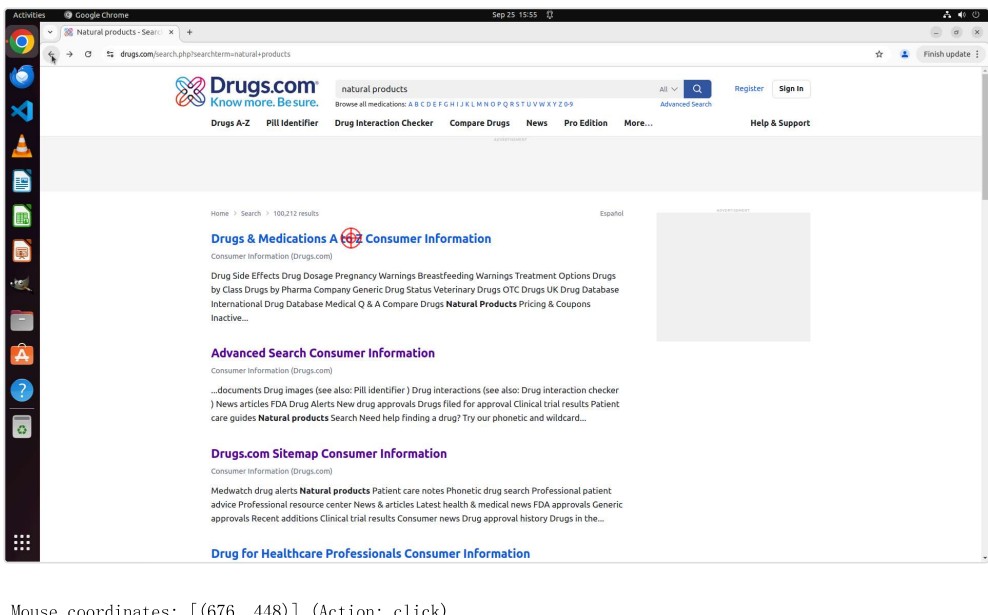

Mouse coordinates: [(676, 448)] (Action: click)

**Assistant:**

**Thought:** I just took a wrong turn and ended up on the advanced search page, but now I've returned to the main search results.
**Explore:** Wait a minute, there's something interesting here – in the second position of the search results list, there's an option called "Drugs & Medications A to Z Consumer Information." Isn't that exactly what we're looking for? Let me click on it and see if this is indeed where we can access detailed information about natural products.
**Memory:** The advance search cannot lead to the natural products database page but another searching page for extra keywords.
**Action:** click(start_box='<|box_start|>(680,453)<|box_end|>')

*Figure 28.* Visualization of EAPO at step 7. With the additional information obtained from the exploration (step 5 and step 6), the agent becomes familiar with this unseen environment and notices the right place to click.

**Step 8**

**System:**

During thinking, you can explore the current page by executing possible actions and memorize the pages you have explored. tags respectively, and the exploration and memory are enclosed within <explore> </explore> and <memory> </memory> tags respectively. For example, <think> This is the reasoning process. </think> <explore> possible actions to explore here </explore> <memory> memory for previous pages here </memory> <think> This is the reasoning process. </think> <action> The final action is \boxed{action here} </action>. In the last part of the action, the final exact action is enclosed within \boxed{} with the action format.

**User:**

You are a GUI agent. You are given a task and your action history, with screenshots. You need to perform the next action to complete the task.

Browse the natural products database.

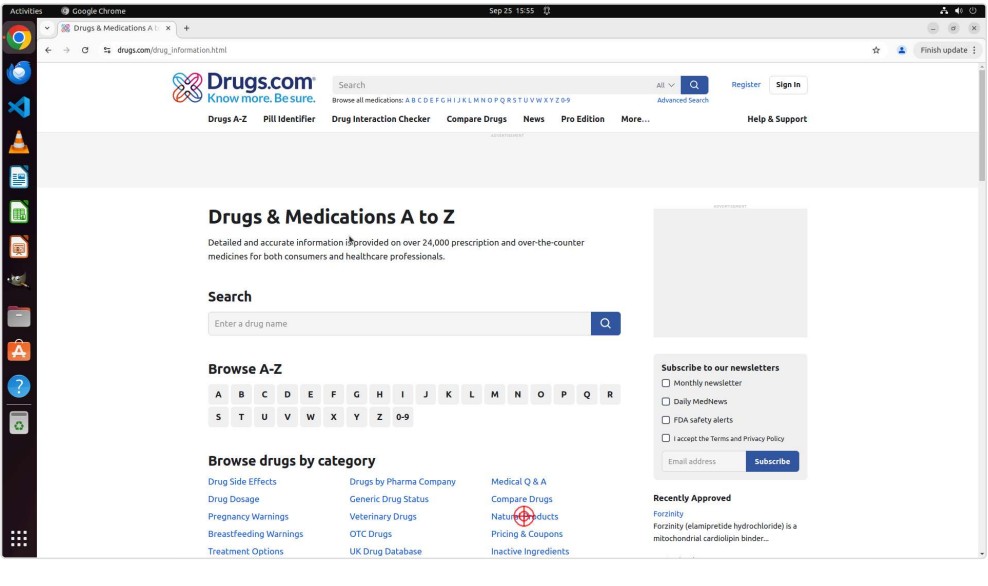

Mouse coordinates: [(1019, 998)] (Action: click)

**Assistant:**

**Thought:** Ah, in the "Browse drugs by category" section, I identified a link labeled "Natural Products," which directly aligns with our objective.
**Explore:** In the "Browse drugs by category" section, I spotted a link labeled "Natural Products," which is exactly what we were searching for. This should be the gateway to our goal, so let me click on it and explore further.
**Memory:** I have tried the advanced search and cannot find further evidence to natural product database. Now I'm back to the main search page.
**Action:** click(start_box='<|box_start|>(1025,1009)<|box_end|>')

*Figure 29.* Visualization of EAPO at step 8.

**Step 9**

**System:**

During thinking, you can explore the current page by executing possible actions and memorize the pages you have explored. tags respectively, and the exploration and memory are enclosed within <explore> </explore> and <memory> </memory> tags respectively. For example, <think> This is the reasoning process. </think> <explore> possible actions to explore here </explore> <memory> memory for previous pages here </memory> <think> This is the reasoning process. </think> <action> The final action is \boxed{action here} </action>. In the last part of the action, the final exact action is enclosed within \boxed{} with the action format.

**User:**

You are a GUI agent. You are given a task and your action history, with screenshots. You need to perform the next action to complete the task.

Browse the natural products database.

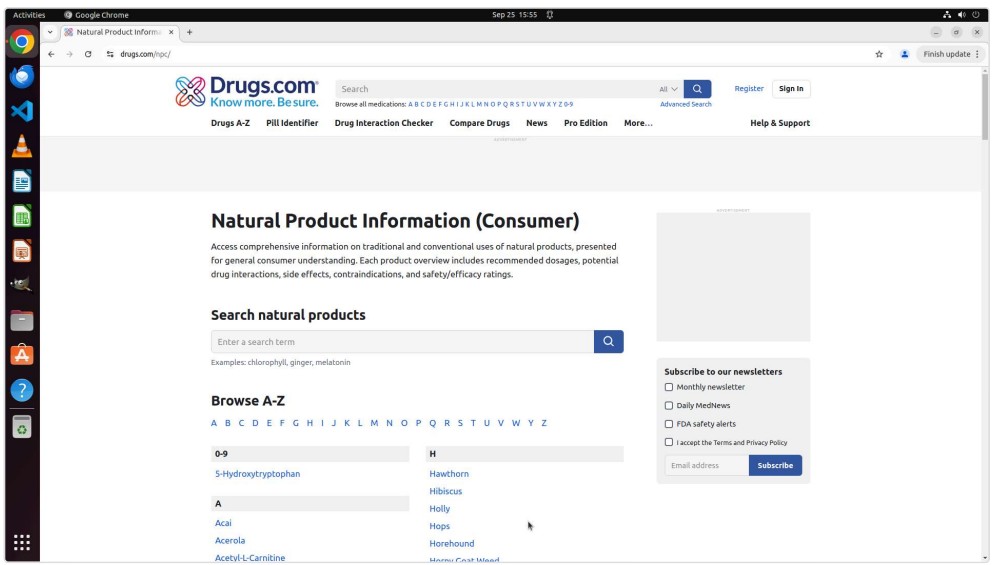

**Assistant:**

**Thought:** That's great! I've finally arrived at the natural products database page. Here, I can find comprehensive information on various natural substances, including their traditional uses, recommended dosages, potential interactions with other medications, and more. The layout of this page is very user-friendly; not only does it feature a search function for quick lookups, but it also offers an alphabetical browsing system that allows users to easily navigate through all the entries from A to Z.
**Action:** finished()

*Figure 30.* Visualization of EAPO at step 9.

