# OpenReview forum: "Learning to Explore: Scaling Agentic Reasoning via Exploration-Aware Policy Optimization"
_ICML.cc/2026/Conference — ICML 2026 regular_

### Official Review · Reviewer_nfrZ · 2026-03-02

**Soundness:** 2
**Presentation:** 3
**Significance:** 2
**Originality:** 2
**Overall Recommendation:** 2
**Confidence:** 4

**Summary:**

This paper proposes EAPO to scale agentic reasoning by teaching agents when to explore vs. act, using explicit exploration + memory outputs and an exploration-aware training objective. It targets interactive text/GUI environments and reports stronger performance than prior prompting- or RL-based baselines under test-time scaling.

**Compliance With Llm Reviewing Policy:**

Affirmed.

**Final Justification:**

I still find that this paper lacks sufficient motivation, a complete and coherent logical flow, and a clear method design. Therefore, I will keep my original score.

**Key Questions For Authors:**

See Weaknesses

**Limitations:**

Yes

**Strengths And Weaknesses:**

Strengths
- The method targets the core failure mode of naive test-time scaling by improving credit assignment for exploratory actions rather than merely increasing sampling or steps.
- The approach is practically oriented for web and GUI agents because it couples structured exploration and memory with a training procedure designed for interactive, feedback-driven settings.



Weaknesses
- The proposed method has over-reliance on rigid output schemas (and format rewards). EAPO hard-codes <explore>/<memory> structure and enforces it with a format reward, which can shift learning toward “looking parseable” rather than doing high-value exploration, and makes the method brittle when the interface, action space, or task format changes. Besides, ReAct already provides an adaptive think–act loop; the motivation for structured exploration guidance and explicit memory is not made decisively clearer, while the explicit tags can also increase token overhead.
- The exploration reward may incentivize plausible behavior rather than truly useful exploration. Because the reward is driven by a learned scoring model, the agent can get reinforced for producing exploration plans and memory notes that look typical or fluent, even if those actions do not actually uncover new task-relevant information or reduce uncertainty in the environment. This risks safer, template-like exploration over targeted, evidence-seeking actions.
- The proposed reward modeling is heuristic and introduces extra tuning burden. It adds additional hyperparameters whose interaction can be sensitive, potentially increasing debugging and training-time iteration cost.
- The grouping heuristic is coarse and fragile in GUI settings. Grouping by (state, visitation depth) ignores the agent’s internal memory/exploration history and depends on reliable “state equality”; both can be noisy in real UIs, injecting variance and bias into advantage estimation.

---

> ### Author Rebuttal · Authors · 2026-03-31
>
> **W1: The proposed method has over-reliance on rigid output schemas (and format rewards).**
>
> To clarify, the structured format does not constrain the semantic content of exploration or memory, but only organizes them into dedicated fields. This separation helps the agent explicitly distinguish between decision-making, information gathering, and state summarization, which facilitates more effective reuse of acquired information and reduces ambiguity during optimization. In particular, the format enables reliable credit assignment for exploration-related rewards, which would otherwise be difficult to achieve with unstructured outputs. Moreover, this output schemas and format rewards are commonly used in recent agentic and reasoning-oriented models, including thinking-style models (Deepseek) and research agents[1]. To corroborate this, we evaluate our method in $4$ doamins with different action space
>
> As directly applying this format may lead to plausible outputs with low quality exploration or over-conservatism (as illustrated in Table.1), we introduce the exploratory reward to evaluate the utility of exploration and guide the agent to gather useful information from the environment for future decision-making.
>
> **W1: Besides, ReAct already provides an adaptive think–act loop.**
>
> Current think–act methods (such as ReAct and ToT) approaches largely rely on static heuristics or predefined branching budgets and lack principled criteria to adaptively control when and how exploration should be conducted during multi-step reasoning. Recent researches have attempted to overcome this limitation by utilizing entropy as an extra signal to balance exploration and exploitation during multi-step reasoning. However, entropy does not faithfully reflect information gain. As far as we know, we are the first to introduce the concept of test-time exploration and train agents to learn to explore at proper time, which allow agents to make attempt in the unknown environment to gather environmental informaton.
>
> **W2: The exploration reward may incentivize plausible behavior rather than truly useful exploration.**
>
> We thank the reviewer for the comment. Although the reward incorporates fine-grained signals, it is not purely heuristic but grounded in estimating the utility of actions in terms of their contribution to successful task completion. When an unintended behaviour is generated, the reward, denoted as $p(e_{t-1},m_{t-1}|s_t,success)$, will be small as the possibility of leading to successful task completion is low. Therefore, the advantage of this action will be negative to punish such wrong action, and only action that helps lead to task completion will be rewarded with high value.
>
> Further, we introduce a discount factor $\gamma$ to prevent over-conservatism, where agents will not meanlessly explore when sufficient information to make a correct decision is obtained.
> These design reduce the risk of the agent exploiting spurious shortcuts that do not contribute to task success.
>
> **W3: The proposed reward modeling introduces extra tuning burden.**
>
> As for training burden, the training curves in Fig.8 show that our method not only exhibits superior performance in terms of task success rate, but also provides a stable performance improvement. Therefore, our method hardly suffers from additional hyperparameters and potentially increasing debugging.
>
> **W4: The grouping heuristic is coarse and fragile in GUI settings.**
>
> In principle, the complete state includes not only the environment state but also the agent’s exploration information and memory, as these jointly determine which action is appropriate. However, grouping transitions by this complete state is infeasible, since exploration histories and memories can differ substantially even when the environment state is identical. To this end, we introduce exploration depth to replace memory and exploration. For actions at the same exploration depth of a state, they should obtain same amount of information during exploration and advantage estimation by grouping these actions are reasonable.
>
> [1] Chen, M., Sun, L., Li, T., Zhu, C., Wang, H., Pan, J.Z., Zhang, W., Chen, H., Yang, F., Zhou, Z. and Chen, W., Learning to Reason with Search for LLMs via Reinforcement Learning. In The Thirty-ninth Annual Conference on Neural Information Processing Systems.

---

> > ### Author Rebuttal · Reviewer_nfrZ · 2026-04-02
> >
> > Thanks for the Rebuttal by Authors. I still believe the proposed method lacks principled algorithm design and the reward modelling is heuristic. The statements like "adaptively control" and "hardly suffers from additional hyperparameters" need clear evidence.

---

> > > ### Author Response · Authors · 2026-04-02
> > >
> > > We sincerely thank the reviewer for this critical point.
> > >
> > > **Q1: The proposed method lacks principled algorithm design and the reward modelling is heuristic.**
> > >
> > > We would like to clarify that our reward model is principled for maximizing task success rate. The reward model is defined as $p(e,m|s,\rm success)$, which characterizes the possibility of a pair of exploration and memory (e,m) successfully completing the task from state $s$. If in all the successful trajectories starting from state $s$, the frequency of the exploration and memory pair is high, then agent which can generate this exploration and memory has a higher possibility of success.
> > > Since the true posterior $p(·|·,\rm success)$ is intractable, we approximate it using a learnable variational proxy with detailed derivation in Appendix. A. This demonstrates that our reward model is equivalent to evaluating exploration and memory by successful task completion.
> > >
> > > **Q2: The statements like "adaptively control" and "hardly suffers from additional hyperparameters" need clear evidence.**
> > >
> > > As illustrated in Fig.15, we verify the exploration degree during training. The exploration degree is defined as the fraction of states that are revisited multiple times among all states, serving as an indicator of the agent’s tendency to explore certain states and revisit them to make the final decision. The results show that the exploration degree, initially close to $0$, gradually increases to a moderate level and then remains stable. This behavior suggests that the agent progressively learns to perform informative exploration while avoiding excessive revisiting, thereby balancing information gathering and decision efficiency.
> > >
> > > Furthermore, as reported in Table 3 in Appendix C, our method adopts standard hyperparameter values that are consistent across different tasks, without requiring task-specific tuning. In addition, the training curves in Fig. 8 demonstrate stable policy improvement over time, while the loss curves in Fig. 12 and Fig. 13 show smooth convergence for both the reward model and the policy model. These empirical results collectively support the claims regarding adaptive behavior and the limited need for additional hyperparameter adjustments.

---

### Official Review · Reviewer_KRv6 · 2026-03-03

**Soundness:** 2
**Presentation:** 3
**Significance:** 2
**Originality:** 2
**Overall Recommendation:** 3
**Confidence:** 4

**Summary:**

This paper investigates a key limitation in agentic test-time scaling methods, i.e., relying on undifferentiated exploration. To solve this, the authors propose an exploration-aware RL framework for LLM agents that triggers exploration only under high uncertainty. The method introduces a fine-grained reward function based on variational inference to explicitly measure the long-term value of exploratory actions, and employs an exploration-aware grouping mechanism to separate exploration from task-completion behaviors during optimization. By focusing exploration on informational gaps and shifting to execution once the context becomes clear, the approach enables more efficient and targeted decision-making. Experiments on challenging text-based and GUI-based agent benchmarks show consistent performance improvements over existing methods.

**Compliance With Llm Reviewing Policy:**

Affirmed.

**Final Justification:**

I thank the authors for their response. However, I still have concerns regarding the paper’s completeness and certain experimental limitations. Therefore, I will maintain my current score.

**Key Questions For Authors:**

**Concern of Efficiency**. The current runtime analysis mainly reports training-time overhead. However, the proposed method relies on adaptive exploration and may incur additional environment interactions, which are not explicitly quantified. For example, with the added rollback actions, what is their cost? It would be helpful to also report and compare rollout-time or interaction costs to give a more complete picture of the computational overhead.

I checked the appendix, page 33, line 1780, where it says, "Further, we demonstrate the time cost of each component, including …", but this sentence seems incomplete. From Fig. 19, rollout time looks dominant. Additionally, the y-axis and the values above each bar don’t seem aligned, or the figure is confused, could the authors clarify this?


**Unclear flow**. I suggest adding a framework figure to clearly illustrate the overall flow of the agent, which would help readers better understand the method. For example, it is unclear when and how the rollback actions are applied during the agent’s decision process.


**Table 1. is not clear**. The authors present the results and refer to "naive exploration", but it is unclear what exactly this entails. Could the authors clarify what kind of naive exploration was used?

**Limitations:**

YES

**Strengths And Weaknesses:**

**Strengths**
1. The paper tackles the challenge of exploration for LLM agents, which is a timely and relevant topic for improving decision-making in agentic RL.

2. The proposed method introduces a fine-grained reward function that explicitly encourages exploration based on information gain, providing a principled way to guide the agent’s exploratory behavior.

3. Experiments on challenging text-based and GUI-based agent benchmarks demonstrate consistent improvements in both performance and convergence speed, supporting the effectiveness of the approach.


**Weaknesses**

1. The reward design is somewhat heuristic. While the fine-grained reward is novel, it is not entirely clear whether it could lead to unintended behaviors, such as reward hacking, especially in more complex environments.

2. Potential additional runtime and token costs are not fully quantified. The method relies on adaptive exploration and rollback actions, which may increase rollout time, environment interactions, and token consumption.

3. Comparisons to exploration-specific baselines are limited. The current baselines appear to focus mainly on task performance rather than exploration. Including comparisons with methods explicitly designed for exploration would strengthen the paper, or the authors could clarify why such baselines are not applicable to the tasks.

---

> ### Author Rebuttal · Authors · 2026-03-31
>
> **W1: The reward design is somewhat heuristic. While the fine-grained reward is novel, it is not entirely clear whether it could lead to unintended behaviors, such as reward hacking, especially in more complex environments.**
>
> We thank the reviewer for the comment. Although the reward incorporates fine-grained signals, it is not purely heuristic but grounded in estimating the utility of actions in terms of their contribution to successful task completion. When an unintended behaviour is generated, the reward, denoted as $p(e_{t-1},m_{t-1}|s_t,\rm success)$, will be small as the possibility of its leading to successful task completion is low. Therefore, the advantage of this action will be negative to punish such wrong action, and only action that helps lead to task completion will be rewarded with high value.
>
> Further, we introduce a discount factor $\gamma$ to prevent over-conservatism, where agents will not meanlessly explore simply for gathering information even when sufficient information to make a correct decision is obtained.
> These design reduce the risk of the agent exploiting spurious shortcuts that do not contribute to task success.
>
> **W2, Q1: Potential additional runtime and token costs are not fully quantified.**
>
> We present comparison of the average step in the following table and observe that our method indeed has additional runtime due to exploration.
> |Method|ReAct|ToT|GRPO|DAPO|GiGPO|LAMER|EAPO(ours)|
> |---|---|---|---|---|---|---|---|
> |AndroidWorld|18.6|24.3|16.9|17.5|19.1|17.2|22.7|
> |OSWorld|21.4|27.8|19.6|20.3|22.0|20.1|21.3|
>
> To avoid high additional runtime by meanless exploration, we introduce introduce a discount factor $\gamma$ so that our method maintains only a modest increase (at most $10\%$) in average step. To be specific, we apply a discount to the exploratory gain since the benefit of exploration is not immediate – requiring at least one step to observe a new state and a subsequent step to synthesize the information
> It guides the agent to carry out exploration only when the anticipated utility ‘outweighs’ the latency cost, which will avoid meanless explortion even when agent has obtained enough information.
>
> **W3: Comparisons to exploration-specific baselines are limited.**
>
> We choose two exploration-specific baselines, ReAct and ToT, which are both inference methods. These methods approaches largely rely on static heuristics or predefined branching budgets and lack principled criteria to adaptively control when and how exploration should be conducted during multi-step reasoning. Recent researches have attempted to overcome this limitation by utilizing entropy as an extra signal to balance exploration and exploitation during multi-step reasoning. However, entropy does not faithfully reflect information gain. As far as we know, we are the first to introduce the concept of test-time exploration and train agents to learn to explore at proper time, which allow agents to make attempt in the unknown environment.
>
> **Q2: I suggest adding a framework figure to clearly illustrate the overall flow of the agent, which would help readers better understand the method.**
>
> We thank the reviewer for the suggestion. We add an overall architecture diagram in https://anonymous.4open.science/r/ICML1847_rebuttal-81D6/framework.png. This framework demonstrates when exploration begins and how the rollback actions are applied during the agent’s decision process.
>
> **Q3: Could the authors clarify what kind of naive exploration was used?**
>
> For the naive exploration in table.1, we directly instruct the different agent to try alternative actions with the prompt present in Appendix C.5, when facing an unfamiliar state until sufficient information is gathered. The results demonstrate that current agents is not capable of exploration by themselves.
>
> **Q1: Additionally, the y-axis and the values above each bar don’t seem aligned, or the figure is confused, could the authors clarify this?**
>
> We complete the sentense as "Further, we demonstrate the time cost of each component, including samplig rollout, calculating old prob, calculating ref prob, updating policy, updating reward, exploration aware grouping and advantage estimation." As the runtime of rollout is much higher than advantage estimation, we break the y-axis for better presentation and the value of time is demonstrated above the bar.

---

> > ### Author Rebuttal · Reviewer_KRv6 · 2026-04-02
> >
> > Thank you for the authors’ response. I would also like to point out several additional baselines that I am aware of:
> >
> > [1] Turning Up the Heat: Min-p Sampling for Creative and Coherent LLM Outputs.
> >
> > [2] Flaming-hot Initiation with Regular Execution Sampling for Large Language Models.
> >
> > [3] Diverse Text Decoding via Iterative Reweighting.
> >
> > I understand that the last work has only recently been accepted. I mention it here considering that this paper has been publicly available for several months.

---

> > > ### Author Response · Authors · 2026-04-02
> > >
> > > We thank the reviewer for the suggestion. We conduct experiments on AndroidWorld and OSWorld with these baselines.
> > >
> > > |Methods|AndroidWorld|OSWorld|
> > > |---|---|---|
> > > |Qwen-VL-2B|46.19|19.08|
> > > |Min-p[1]|50.82|23.11|
> > > |FIRE[2]|53.35|23.46|
> > > |OverRIDE[3]|51.07|24.02|
> > > |GiGPO|56.01|28.71|
> > > |LAMER|61.26|33.76|
> > > |EAPO (ours)|76.36|50.34|
> > >
> > > As illustrated in the table, these methods demonstrate no superiority compared with RL methods and we summarize the following two reasons. First, they are designed for LLM tasks and aim to increase diversity within a single generation, whereas agentic tasks require interaction with the environment. Such sampling strategies can only produce more diverse actions at the current state, but cannot obtain new information through environment transitions to improve subsequent decisions. Second, these methods control exploration using heuristics such as temperature or entropy, which do not explicitly optimize task success. In contrast, our method learns a principled reward that evaluates the utility of exploration and trains the agent to actively explore informative states, enabling it to acquire environmental information at test time to support decision-making.
> > >
> > > [1] Turning Up the Heat: Min-p Sampling for Creative and Coherent LLM Outputs.
> > >
> > > [2] Flaming-hot Initiation with Regular Execution Sampling for Large Language Models.
> > >
> > > [3] Diverse Text Decoding via Iterative Reweighting.

---

### Official Review · Reviewer_nehj · 2026-03-12

**Soundness:** 3
**Presentation:** 2
**Significance:** 2
**Originality:** 4
**Overall Recommendation:** 4
**Confidence:** 3

**Summary:**

This paper proposes the Exploration-Aware Policy Optimization (EAPO) framework, aimed at addressing the exploration efficiency problem of LLM agents during test-time scaling. Existing methods tend to be conservative or blind due to their lack of ability to execute valuable actions and integrate exploratory information. EAPO introduces an explicit "exploration and memory" reasoning paradigm, trains a reward function using variational inference to evaluate the utility of exploratory actions, and designs a two-stage training strategy incorporating SFT rollback and exploration-aware grouping. Evaluations across multiple environments demonstrate that EAPO significantly outperforms existing methods in decision-making capability and exhibits strong generalization.

**Compliance With Llm Reviewing Policy:**

Affirmed.

**Final Justification:**

My concerns have been fully resolved by the authors' rebuttal.

**Key Questions For Authors:**

As shown in Table 7 of the paper, after removing the SFT module (w/o SFT), EAPO's performance on multiple benchmarks (e.g., ALFWorld, WebShop) drops sharply (for example, Qwen-VL-8B on ALFWorld decreases from 76.0 to 48.0, a drop of 26.0). Moreover, its performance at this point falls below or is on par with existing advanced methods such as GRPO and DAPO (Table 2). This indicates that EAPO's performance improvement does not primarily stem from its proposed "access depth-based policy optimization" or new reward mechanism, but rather relies heavily on the rollback mechanism data injected during the SFT phase. If the method loses its competitiveness after removing SFT, then the core innovation of this work appears to be more about "data distillation using a strong teacher model" rather than improvements to the algorithmic architecture itself, which casts doubt on the effectiveness and independence of EAPO as a general reinforcement learning framework.

1.could you provide more details on the selection criteria for the teacher LLM and the validation mechanisms used during the rollback data collection to ensure the rationality and robustness of the generated data, given this heavy reliance?
2.If the method loses its competitiveness without SFT, how do you justify the effectiveness and independence of EAPO as a reinforcement learning framework?
See the Weaknesses section for the rest.

**Limitations:**

The primary limitation of this work lies in the heavy dependency on the quality and capabilities of the teacher model for generating rollback data; the ablation study (Table 7) indicates that without the SFT initialization using this teacher-generated data, the proposed EAPO framework loses its competitive edge against standard baselines, raising concerns about whether the performance gains stem from the algorithmic novelty or merely from high-quality data distillation. Secondly, the empirical validation is currently restricted to embodied and GUI interaction tasks, leaving the generalization capability of EAPO to symbolic reasoning, code generation, or mathematical domains unverified. Furthermore, the paper lacks a holistic architectural diagram to intuitively illustrate the interplay between the reasoning paradigm, reward modeling, and the two-stage training strategy, which hinders immediate comprehension of the system's workflow. Finally, the absence of detailed inference hyperparameters (e.g., temperature, max tokens, sampling strategies) and specific engineering stacks limits the reproducibility of the results and the fairness of comparisons with existing methods.

**Strengths And Weaknesses:**

Strengths:
1. The paper provides a profound insight into the critical disconnect between reward modeling and policy training, explicitly identifying that even with accurate reward models, standard training pipelines still face fundamental challenges such as attribution bias, lack of rollback mechanisms, and optimization objective confusion. Based on this diagnosis, the paper proposes the EAPO two-stage training algorithm, which specifically addresses the core pain point of underestimated utility of exploratory actions, demonstrating methodological innovation rooted in problem essence.

2. The paper demonstrates solid theoretical depth in reward modeling, not only constructing an accurate reward function but also delving into the fundamental disconnect between "accurate rewards" and "effective training"—pointing out that even with reasonable rewards, standard pipelines struggle to unleash the value of exploration due to issues like attribution bias and lack of rollback mechanisms. This complete path from theoretical diagnosis to algorithmic design reflects the rigor of methodological innovation.

3. Effectiveness is validated across multiple types of embodied tasks and GUI environments, with experimental results comprehensively outperforming mainstream baselines. The ablation studies are thorough, verifying the contribution of each component one by one, presenting a rigorous chain of evidence. This provides a solid foundation for subsequent research in the field.

Weaknesses:

1. The paper currently lacks an overarching architecture diagram. Readers need to mentally piece together the connections between the "reasoning paradigm (Section 4.1)", "reward modeling (Section 4.3)", and "training algorithm (Section 5)". The current illustrations focus mainly on experimental results, leaving readers without an intuitive "big picture" when reading the main text, making it difficult to grasp at a glance how this complex multi-module system works collaboratively.

2. The experiments are limited to GUI and embodied tasks, and do not validate effectiveness in domains such as reasoning, code, or mathematics.

3. The paper does not clearly report key hyperparameters and technical stacks for different methods during the inference phase, such as temperature, maximum generation length (max tokens), whether sampling strategies are enabled, and the inference frameworks used (e.g., vLLM, SGLang, etc.). Such information is crucial for result reproducibility and fair comparison, and its absence diminishes the transparency and engineering reference value of the experiments.

4. There is excessive reliance on teacher model capabilities and unclear attribution of core contributions: the paper does not specify the details (parameter size, training corpus) of the "teacher LLM". If the teacher model itself does not possess perfect environment reasoning capabilities, its generated rollback actions may contain hidden logical errors or hallucinations.

---

> ### Author Rebuttal · Authors · 2026-03-31
>
> **W1: The paper currently lacks an overarching architecture diagram.**
>
> We thank the reviewer for the suggestion. We add an overall architecture diagram in https://anonymous.4open.science/r/ICML1847_rebuttal-81D6/framework.png. This framework demonstrates when exploration begins, how the rollback actions are applied during the agent’s decision process, how rollback data is collected, and how the agent is trained to explore.
>
> **W2: The experiments are limited to GUI and embodied tasks, and do not validate effectiveness in domains such as reasoning, code, or mathematics.**
>
> Our experiments focus on GUI and embodied environments because they are inherently agentic tasks that require active interaction with external environments. In contrast, domains such as reasoning, code, and mathematics are typically non-agentic settings, where the model operates in a static context without the need for environment interaction or information gathering through exploration. Since our method is specifically designed to learn when to explore and how to acquire useful information via interaction, GUI and embodied tasks provide more appropriate testbeds for evaluating its effectiveness.
>
> **W3: The paper does not clearly report key hyperparameters and technical stacks for different methods during the inference phase.**
>
> Model deployment employs vLLM as the rollout service and the sampling temperature is set to $1.0$, with a maximum generation length of $4096$ tokens.
>
> **W4, Q2: Could you provide more details on the selection criteria for the teacher LLM and the validation mechanisms used during the rollback data collection to ensure the rationality and robustness of the generated data, given this heavy reliance.**
>
> We first collect forward transitions $(s,a,s')$. Then we prompt a stronger model (Deepseek-R1-0528) with the state $s'$ and instruction to generate rollback action $a$. We execute this action at state $s'$ to obtain rollback transition $(s',a,s'')$. If the transit state $s''$ is the same as the original state $s$, then we collect this rollback transition $(s',a,s)$. If this model's generated rollback actions may contain hidden logical errors or hallucinations, it cannot transit to the correct previous state $s$, which transition wll not be selected as the data for SFT.
>
> **Q1: If the method loses its competitiveness without SFT, how do you justify the effectiveness and independence of EAPO as a reinforcement learning framework? **
>
> First, our ablation studies in Tab.7 show that models trained with SFT alone still perform poorly, indicating that SFT is not the primary factor driving the performance gains. In contrast, significant improvements are only observed when reinforcement learning with exploratory reward is applied, demonstrating the effectiveness of EAPO as an RL-based optimization framework.
>
> Second, exploration accuracy and rollback capability are inherently coupled. Even if the agent learns to explore effectively, the acquired information cannot be properly utilized without the ability to rollback to previous decision points. As a result, removing the SFT stage—which provides initial rollback capability—leads to a noticeable performance drop. This does not imply that SFT is responsible for the final performance, but rather that it serves as an enabling component for effective exploration-aware learning.
>
> Third, the SFT data used in our method focuses on learning rollback behaviors and does not include forward decision-making supervision. Therefore, distilling only rollback trajectories is insufficient to solve the task.

---

> > ### Author Rebuttal · Reviewer_nehj · 2026-04-03
> >
> > My concern has been resolved. I hope you can add this point in the final version of the paper, and I am willing to raise my score.

---

> > > ### Author Response · Authors · 2026-04-03
> > >
> > > We thank the reviewer for the insightful comment. We would like to clarify that our method is not intended to be a standalone RL framework, but rather a training approach that enables agents to adaptively decide when and what to explore. Achieving this goal requires two key capabilities. First, the agent must be able to evaluate the usefulness of information obtained through exploration. To this end, we design a reward model together with exploration-aware group advantage estimation, which allows the policy to learn the utility of exploration for downstream decision-making. Second, the agent must be able to accurately return to the pre-exploration state so that the acquired information can be effectively utilized. For this purpose, we introduce rollback data generation and use SFT to bootstrap this capability.
> > >
> > > Importantly, the SFT stage is not responsible for improving task performance. To verify this, we initialize the model with the same SFT checkpoint and then train it using baseline methods.
> > >
> > > |Method|AndroidWorld|OSWorld|
> > > |---|---|---|
> > > |Qwen-VL-2B|46.19|19.08|
> > > |Qwen-VL-2B (SFT)|47.03|19.62|
> > > |GRPO|52.38|22.87|
> > > |GRPO (SFT)|53.01|23.45|
> > > |DAPO|56.82|25.18|
> > > |DAPO (SFT)|57.34|25.73|
> > > |GiGPO|56.01|28.71|
> > > |GiGPO (SFT)|56.88|29.12|
> > > |LAMER|61.26|33.76|
> > > |LAMER (SFT)|62.05|34.28|
> > > |ours|76.36|50.34|
> > >
> > > The results in the above table indicate that models with SFT initialization still perform poorly, demonstrating that the teacher model does not directly provide performance gains. Instead, the improvements arise only when the exploration-aware reward and optimization strategy are applied, confirming that the effectiveness primarily stems from our adaptive exploration learning mechanism rather than knowledge inherited from the teacher.

---

### Official Review · Reviewer_1Lx2 · 2026-03-13

**Soundness:** 3
**Presentation:** 3
**Significance:** 3
**Originality:** 3
**Overall Recommendation:** 5
**Confidence:** 3

**Summary:**

This paper proposes EAPO as an RL framework that teaches LLM agents to explore adaptively. This is done through introducing <explore> and <memory> tags to surface exploration and memory in text, and a variational reward that credits informative exploratory actions, and exploration-aware grouping for stable optimization. Results show strong gains across agentic benchmarks, with comprehensive analysis and ablations.

**Compliance With Llm Reviewing Policy:**

Affirmed.

**Final Justification:**

My main concerns were addressed with additional results so I have no significant remaining concerns

**Key Questions For Authors:**

See weaknesses

**Limitations:**

yes

**Strengths And Weaknesses:**

Strengths
- Variational reward is a neat way to credit exploration without online rollouts
- Strong results throughout, with 2B model beating much larger models on AndroidWorld/OSWorld is impressive
- Good benchmark coverage (4 envs, 3 scales, 6 baselines)
- Exploration-aware grouping and sft rollback are well-motivated, and supported by ablations
- The paper is well-written

Weaknesses
- The exploratory reward itself is not ablated, and it is possible the rollback + grouping doing the heavy lifting
- I'm not entirely sure baseline comparisons are fair. It seems EAPO agents get extra environment steps for exploration and rollback? It would be good if the authors can clarify.

---

> ### Author Rebuttal · Authors · 2026-03-31
>
> **W1: The exploratory reward itself is not ablated, and it is possible the rollback + grouping doing the heavy lifting**
>
> To verify the neccessity of exploratory reward, we conduct experiment on ablating the exploratory reward and present the results in the following table.
> |Method|ALFworlf|WebShop|AndroidWorld|OSWorld|
> |---|---|---|---|---|
> |w/o exploratory reward|37.6|30.3|51.0|23.7|
> |EAPO|58.5|53.2|76.3|50.3|
>
> Without exploratory reward, agents fail to evaluate the usefulness of each exploratory action, causing ineffective exploration and premature convergence to suboptimal behaviors, which leads to significantly degraded performance across all environments.
>
> **W2: It seems EAPO agents get extra environment steps for exploration and rollback?**
>
> We present comparison of the average step in the following table and observe that our method indeed has additional runtime due to exploration.
> |Method|ReAct|ToT|GRPO|DAPO|GiGPO|LAMER|EAPO(ours)|
> |---|---|---|---|---|---|---|---|
> |AndroidWorld|18.6|24.3|16.9|17.5|19.1|17.2|22.7|
> |OSWorld|21.4|27.8|19.6|20.3|22.0|20.1|21.3|
>
> To avoid meanless exploration, we introduce introduce a discount factor $\gamma$ so that our method maintains only a modest increase (at most $10\%$) in average step. To be specific, we apply a discount to the exploratory gain since the benefit of exploration is not immediate – requiring at least one step to observe a new state and a subsequent step to synthesize the information
> It guides the agent to carry out exploration only when the anticipated utility ‘outweighs’ the latency cost, which will avoid meanless explortion even when agent has obtained enough information.

---

> > ### Author Rebuttal · Reviewer_1Lx2 · 2026-04-06
> >
> > The authors provided additional results regarding my concerns.

---

> > > ### Author Response · Authors · 2026-04-06
> > >
> > > Thank you so much for your timely response and support for acceptance!

---

### Decision · Program_Chairs · 2026-04-30

**Decision:**

Accept (regular)

**Comment:**

The core justification for acceptance is the synergy hypothesis (between SFT rollback and the exploration component) demonstrated in the rebuttal. The authors provided empirical evidence that standard RL baselines (GRPO, LAMER) do not improve significantly when initialized with the same high-quality SFT rollback data. This seems to indicate that while the teacher model provides the capability to backtrack, EAPO provides the intelligence to know when and what to explore. The empirical result of a 2B model rivaling 70B frontier models on OSWorld is a major contribution to efficient agentic scaling that outweighs the concerns regarding the heuristic nature of the output tags.

Main concern remaining: The significant performance collapse observed in the absence of SFT rollback suggests that the framework lacks algorithmic independence and may not yet be a generalized solution for RL exploration, outside GUI environments and in the absence of a strong model.